# Enhancing Sample Generation of Diffusion Models using Noise Level Correction

**Abulikemu Abuduweili**                                        *abulikea@andrew.cmu.edu*
*Carnegie Mellon University*

**Chenyang Yuan**                                              *chenyang.yuan@tri.global*
*Toyota Research Institute*

**Changliu Liu**                                                  *cliu6@andrew.cmu.edu*
*Carnegie Mellon University*

**Frank Permenter**                                          *frank.permenter@tri.global*
*Toyota Research Institute*

**Reviewed on OpenReview:** *https://openreview.net/forum?id=y8VXikiIUO*

## Abstract

The denoising process of diffusion models can be interpreted as an approximate projection of noisy samples onto the data manifold. Moreover, the noise level in these samples approximates their distance to the underlying manifold. Building on this insight, we propose a novel method to enhance sample generation by aligning the estimated noise level with the true distance of noisy samples to the manifold. Specifically, we introduce a noise level correction network, leveraging a pre-trained denoising network, to refine noise level estimates during the denoising process. Additionally, we extend this approach to various image restoration tasks by integrating task-specific constraints, including inpainting, deblurring, super-resolution, colorization, and compressive sensing. Experimental results demonstrate that our method significantly improves sample quality in both unconstrained and constrained generation scenarios. Notably, the proposed noise level correction framework is compatible with existing denoising schedulers (e.g., DDIM), offering additional performance improvements.

## 1 Introduction

Generative models have significantly advanced our capability of creating high-fidelity data samples across various domains such as images, audio, and text (Song & Ermon, 2019; Sohl-Dickstein et al., 2015; Oussidi & Elhassouny, 2018). Among these, diffusion models have emerged as one of the most powerful approaches due to their superior performance in generating high-quality samples from complex distributions (Song & Ermon, 2019; Sohl-Dickstein et al., 2015; Li et al., 2024a). Unlike previous generative models, such as generative adversarial networks (GANs) (Goodfellow et al., 2014) and variational autoencoders (VAEs) (Kingma & Welling, 2014), diffusion models add multiple levels of noise to the data, and the original data is recovered through a learned denoising process (Ho et al., 2020; Song et al., 2023). This allows diffusion models to handle high-dimensional, complex data distributions, making them especially useful for tasks where sample quality and diversity are critical (Cao et al., 2024). Their capability to generate complex, high-resolution data has led to widespread applications across numerous tasks, from image generation in models like DALL · E (Ramesh et al., 2022) and Stable Diffusion (Rombach et al., 2022) to use in robotic path-planning and control (Janner et al., 2022; Chi et al., 2023), as well as text generation (Li et al., 2022).

Previous studies (Rick Chang et al., 2017; Permenter & Yuan, 2024) have interpreted the denoising process in diffusion models as an approximate projection onto the data manifold, with the noise level $\sigma_t$ approximating

the distance between noisy samples and the data manifold. This perspective views the sampling process as an optimization problem, where the goal is to minimize the distance between noisy samples and the underlying data manifold using gradient descent. The gradient direction is approximated by the denoiser output with step size determined by the noise level schedule. However, the denoiser requires an estimate of the noise level as input. We claim that accurately estimating the noise level during the denoising process—essentially the distance to the data manifold—is crucial for convergence and accurate sampling.

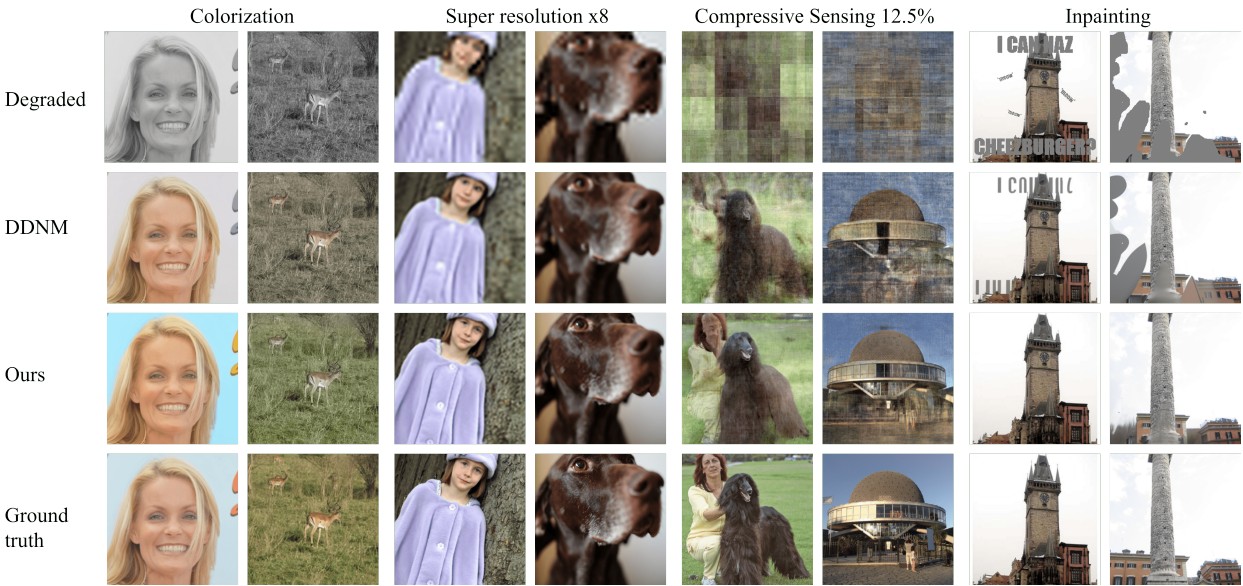

Figure 1: Qualitative results of constrained image generation.

The expressive capabilities of diffusion models have also made them a compelling choice for image restoration tasks, where generating high-quality, detailed images is essential (Wang et al., 2024). Diffusion models can be used as an image-prior for capturing the underlying structure of image manifold and have shown significant promise for constrained generation, such as image restoration (Song et al., 2022; Chung et al., 2023a; Wu et al., 2024). Plug-and-play methods were proposed to utilize pre-trained models without the need for extensive retraining or end-to-end optimization for linear inverse problems such as super-resolution, inpainting, and compressive sensing (Chung et al., 2023a; Dou & Song, 2024; Wang et al., 2023). These methods can be interpreted as alternating taking gradient steps towards the constraint set (linear projection for linear inverse problems) and the image manifold (noise direction estimated by the learned denoiser) to find their intersection. However, they may suffer from inconsistency issues if, after each projection step, the noise level no longer approximates distance (Figure 2). Thus, correcting the noise level after each step could increase the accuracy of image restoration tasks.

In this work, we propose a novel noise level correction method to refine the estimated noise level and enhance sample generation quality. Our approach introduces a noise level correction network that aligns the estimated noise level of noisy samples more closely with their true distance to the data manifold. By dynamically adjusting the sampling step size based on this corrected noise level estimation, our method improves the sample generation process, significantly enhancing the quality of generated data. Furthermore, our approach integrates seamlessly with existing denoising scheduling methods, such as DDPM (Denoising Diffusion Probabilistic Models) (Ho et al., 2020), DDIM (Denoising Diffusion Implicit Models) (Song et al., 2021a), and EDM (Karras et al., 2022). Furthermore, we extend the application of noise level correction to various image restoration tasks, showing its ability to improve the performance of diffusion-based models such as DDNM (Denoising Diffusion Nullspace Model) by (Wang et al., 2023). Our method achieves improved results across tasks including inpainting, deblurring, super-resolution, colorization, and compressive sensing, as illustrated in Figure 1. Additionally, we introduce a parameter-free lookup table as an approximation of the noise level correction network, providing a computationally efficient alternative for improving the performance of unconstrained diffusion models. In summary, our contributions are:

- We propose a noise level correction network that improves sample generation quality by dynamically refining the estimated noise level during the denoising process.
- We extend the proposed method to constrained tasks, achieving significant performance improvements in various image restoration challenges.
- We develop a parameter-free approximation of the noise level correction network, offering a computationally efficient tool to improve diffusion models.
- Through extensive experiments, we demonstrate that the proposed noise level correction method consistently provides additional performance gains when applied on top of various denoising methods. Our code is available at `https://github.com/Walleclipse/Diffusion-NLC/`.

## 2 Background

### 2.1 Diffusion Models

Diffusion models represent a powerful class of latent variable generative models that treat datasets as samples from a probability distribution, typically assumed to lie on a low-dimensional manifold $\mathcal{K} \subset \mathbb{R}^n$ (Sohl-Dickstein et al., 2015; Ho et al., 2020). Given a data point $z_0 \in \mathcal{K}$, diffusion models aim to learn a model distribution $p_\theta(z_0)$ that can approximate this manifold and enable the generation of high-quality samples. The process involves gradually corrupting the data with noise during a forward diffusion process and incrementally denoising it to reconstruct the original data through a reverse generative process.

**Diffusion (forward) process.** In the forward process, noise is added progressively to the data. Starting with a clean sample $z_0$, the noisy version at step $t$, denoted $z_t$, is a linear combination of $z_0$ and Gaussian noise $\epsilon$:

$$z_t = \sqrt{\alpha_t} z_0 + \sqrt{1 - \alpha_t} \epsilon, \tag{1}$$

where $\epsilon \sim \mathcal{N}(0, I)$ and the noise schedule $\alpha_t$ controls the amount of noise injected at each step. Typically, $1 > \alpha_1 > \alpha_2 > \cdots > \alpha_T > 0$, ensuring that $p(z_T) \approx \mathcal{N}(0, I)$ for large enough $T$ (Ho et al., 2020). For mathematical convenience, a reparameterization is often employed, defining new variables $x_t = z_t / \sqrt{\alpha_t}$, which results in:

$$x_t = x_0 + \sigma_t \epsilon, \text{ where } \epsilon \sim \mathcal{N}(0, I), \tag{2}$$

$$\sigma_t = \sqrt{\frac{1 - \alpha_t}{\alpha_t}}, \ x_t = \frac{z_t}{\sqrt{\alpha_t}}, \ x_0 = z_0. \tag{3}$$

Where $\sigma_t$ denotes the noise level. Note that the diffusion process is originally presented in variable $z_t$, we use the formulation $x_t = \frac{z_t}{\sqrt{\alpha_t}}$ to simplify the forward and reverse denoising processes. A similar formulation can be found in (Song et al., 2021b; Karras et al., 2022).

**Denoiser.** Diffusion models are trained to estimate the noise vector added to a sample during the forward process. The learned denoiser, denoted as $\epsilon_\theta$, is to predict the noise vector $\epsilon$ from the noisy sample $x_t$ and the corresponding noise level $\sigma_t$. The denoiser is optimized using a loss function that minimizes the difference between the predicted and true noise vectors:

$$L(\theta) := \mathbf{E} \|\epsilon_\theta(x_t, \sigma_t) - \epsilon\|^2 = \mathbf{E}_{x_0, t, \epsilon} \|\epsilon_\theta(x_0 + \sigma_t \epsilon, \sigma_t) - \epsilon\|^2. \tag{4}$$

Here, $x_0$ is sampled from the data distribution, $\sigma_t$ drawn from a discrete predefined noise level schedule, and $\epsilon$ is drawn from a standard Gaussian distribution, $\mathcal{N}(0, I)$. Training is typically performed using gradient descent, where randomly sampled triplets $(x_0, \epsilon, \sigma_t)$ are used to update the denoiser's parameters $\theta$. Once the denoiser is trained, we can apply a one-step estimation to approximate the clean sample $\hat{x}_{0|t} \approx x_0$:

$$\hat{x}_{0|t} = x_t - \sigma_t \epsilon_\theta(x_t, \sigma_t). \tag{5}$$

**Denoising (sampling) process.** The one-step estimation, eq. (5), may lack accuracy, in which case the trained denoiser is applied iteratively through the denoising process. This process aims to progressively

denoise a noisy sample $x_T$ and recover the original data $x_0$. Sampling algorithms construct a sequence of intermediate estimates $(x_T, x_{T-1}, \ldots, x_0)$, starting from an initial point $x_T$ drawn from a Gaussian distribution, $x_T \sim \mathcal{N}(0, I)$. One of the widely used samplers, the deterministic DDIM (Song et al., 2021a), follows the recursion:

$$x_{t-1} = x_t + (\sigma_{t-1} - \sigma_t)\epsilon_\theta(x_t, \sigma_t) = \hat{x}_{0|t} + \sigma_{t-1}\epsilon_\theta(x_t, \sigma_t) \tag{6}$$

$$x_T = \frac{z_T}{\sqrt{\alpha_t}} = \sqrt{\sigma_T^2 + 1} \cdot z_T, \ z_T \sim \mathcal{N}(0, I), \tag{7}$$

where $\epsilon_\theta$ is the predicted noise at step $t$. This iterative process continues until $x_0$ is obtained, which represents a denoised sample. On the other hand, the randomized DDPM (Ho et al., 2020) follows the update rule:

$$x_{t-1} = x_t + (\sigma_{t'} - \sigma_t)\epsilon_\theta(x_t, \sigma_t) + \eta\omega_t = \hat{x}_{0|t} + \sigma_{t'}\epsilon_\theta(x_t, \sigma_t) + \eta\omega_t \tag{8}$$

$$\sigma_{t'} = \frac{\sigma_{t-1}^2}{\sigma_t}, \ \eta = \sqrt{\sigma_{t-1}^2 - \sigma_{t'}^2}, \ \omega_t \sim \mathcal{N}(0, I). \tag{9}$$

## 2.2 Additional Related Works

Diffusion models have gained significant attention for their ability to learn complex data distributions, excelling in diverse applications such as image generation (Ho et al., 2020; Arechiga et al., 2023), audio synthesis (Kong et al., 2021), and robotics (Chi et al., 2023). On the theoretical side, significant research has explored the non-asymptotic convergence rates of various diffusion samplers, including DDPM (Benton et al., 2024) and DDIM (Li et al., 2024b), contributing to a deeper understanding of the optimization processes underlying diffusion-based models.

**Few-step Sampling.** Several methods have been proposed to improve the quality of generated samples while reducing the number of iterations. The first class of these techniques require modifications to the training process, or training of additional models. They include model distillation (Nichol & Dhariwal, 2021), progressive distillation (Salimans & Ho, 2022) or using consistency models (Song et al., 2023; Kim et al., 2023) to predict the endpoint of the sampling process. They can usually produce samples in fewer than 10 steps but are computationally expensive to train and could suffer from training instabilities. The second class of techniques do not require any additional training but instead modifies the sampling process. These include faster samplers known as EDM (Karras et al., 2022), DPM (Lu et al., 2022) or modifications to the sampling noise schedule as in (Sabour et al., 2024; Hang et al., 2024). They do not need additional training and can be directly used on pre-trained models, but this also limits their performance compared to training-based methods.

**Adaptive noise schedules.** Our proposed method can be seen as a way of introducing additional computation in the sampling process to adaptively change the sampling noise schedule depending on input images. Noise schedules play an important role during sampling, and (Chen, 2023) showed that different schedules are optimal for different data. To tailor the noise schedule to different inputs, (Sahoo et al., 2024) learns to adapt the noise schedule depending on both the pixel location and input. AdaDiff (Tang et al., 2024) dynamically adjusts the inference computation during sampling using a learned uncertainty estimation module.

**Diffusion models for inverse problems.** Diffusion models have also demonstrated effectiveness in image restoration tasks, including super-resolution (Chung et al., 2023a), inpainting (Kawar et al., 2022), deblurring, and compressive sensing (Wang et al., 2023). These tasks often require strict adherence to data consistency constraints, making diffusion models particularly suitable for addressing such challenges. Prior work such as DDRM (Kawar et al., 2022) and DDNM (Wang et al., 2023) pioneered the use of diffusion models to solve linear inverse problems by projecting noisy samples onto the subspace of solutions that satisfy linear constraints. Other methods have employed alternative approaches, such as computing full projections by enforcing linear constraints or using the gradient of quadratically penalized constraints (Chung et al., 2022; 2023b). Recent work has extended the application of diffusion models to more complex non-convex constraint functions. In these cases, iterative methods such as gradient descent are utilized to guide the sampling process toward satisfying the constraints, as demonstrated in Universal Guidance (Bansal et al., 2023). Furthermore, a provably robust framework for score-based diffusion models applied to image reconstruction

was introduced by (Xu & Chi, 2024), offering robust performance in handling nonlinear inverse problems while ensuring consistency with the observed data.

## 3 Methods

### 3.1 Noise Level as Distance from Noisy Samples to the Manifold

This work builds on the insight that denoising in diffusion models can be interpreted as an approximate projection onto the support of the training-set distribution. Previous studies (Rick Chang et al., 2017; Permenter & Yuan, 2024) have established this connection. The distance function of a set $\mathcal{K} \subset \mathbb{R}^n$, denoted as $\text{dist}_\mathcal{K}(x)$, is defined as the minimum distance from a point $x$ to any point $x_0 \in \mathcal{K}$:

$$\text{dist}_\mathcal{K}(x) := \inf\{\|x - x_0\| : x_0 \in \mathcal{K}\}. \tag{10}$$

The projection of $x$ onto $\mathcal{K}$, denoted $\text{proj}_\mathcal{K}(x)$, refers to the point (or points) on $\mathcal{K}$ that achieves this minimum distance. Assuming the projection is unique, we can express it as:

$$\text{proj}_\mathcal{K}(x) := \{x_0 \in \mathcal{K} : \text{dist}_\mathcal{K}(x) = \|x - x_0\|\}. \tag{11}$$

(Permenter & Yuan, 2024) introduce the following relative-error model, inspired by the manifold hypothesis (Bengio et al., 2013; Fefferman et al., 2016) as well as from the fact that the ideal/optimal denoiser is the gradient of a smoothed distance function:

**Assumption 3.1 (Informal)** *The learned denoiser $\epsilon^*$ as well as the sampling noise schedule $\{\sigma_t\}_{t=1}^N$ satisfies the following for all $x_t$ during sampling:*

1. *$\sigma_t$ decreases slowly enough to ensure that $\sqrt{n}\sigma_t \approx \text{dist}_\mathcal{K}(x_t)$ after each iteration*

2. *If $\sigma_t$ approximates $\text{dist}_\mathcal{K}(x_t)$, then $x_t - \sigma_t\epsilon^*(x_t, \sigma_t) \approx \text{proj}_\mathcal{K}(x_t)$ with constant relative error with respect to $\text{dist}_\mathcal{K}(x_t)$.*

Leveraging this assumption, we outline the following theorem informally, which summarizes results from (Permenter & Yuan, 2024) (for formal proofs refer to (Permenter & Yuan, 2024)):

**Theorem 3.1 (Informal)** *Suppose Assumption 3.1 holds and the initial distance satisfies $\text{dist}_\mathcal{K}(x_T) = \sqrt{n}\sigma_T$. Then the DDIM sampler generates the sequence $(x_T, \ldots, x_0)$ by performing gradient descent on the objective function $f(x) := \frac{1}{2}\text{dist}_\mathcal{K}(x)^2$ with a step-size of $\beta_t := 1 - \sigma_{t-1}/\sigma_t$:*

$$x_{t-1} = x_t - \beta_t \nabla f(x_t) = x_t - \beta_t \cdot \text{dist}_\mathcal{K}(x_t) \cdot \nabla \text{dist}_\mathcal{K}(x_t), \tag{12}$$

$$\text{dist}_\mathcal{K}(x_t) = \sqrt{n}\sigma_t, \quad \nabla \text{dist}_\mathcal{K}(x_t) = \epsilon_\theta^*(x_t, \sigma_t)/\sqrt{n} \tag{13}$$

Theorem 3.1 demonstrates that the estimated clean sample generated by the denoising process $\hat{x}_{0|t} = x_t - \sigma_t\epsilon_\theta(x_t, \sigma_t))$ serves as an approximation of the projection of the noisy sample $x_t$ onto the manifold $\mathcal{K}$. It also establishes that throughout the sampling process, the distance of a noisy sample $x_t$ onto manifold $\mathcal{K}$, can be approximated by the noise level $\sqrt{n}\sigma_t$. Given that $\sigma_t$ and $\epsilon_\theta$ satisfies Assumption 3.1, the DDIM process generates a sequence $d_t$ with monotonically decreasing distance to the manifold by gradient descent on the squared-distance function to the manifold. Specifically, the distance of each noisy sample from the manifold is determined by the noise level, $\text{dist}_\mathcal{K}(x_t) = \sqrt{n}\sigma_t$, while the gradient $\nabla f(x)$, which is the direction of projection of $x_t$ onto the manifold, is estimated by the noise vector $\nabla \text{dist}_\mathcal{K}(x_t) \approx \epsilon_\theta^*(x_t, \sigma_t)/\sqrt{n}$.

Theorem 3.1 exposes two key assumptions that should be satisfied by the denoising process in order to obtain good samples on $\mathcal{K}$: (1) a denoiser $\epsilon_\theta(x_t, \sigma_t)$ satisfying the relative projection-error assumption, and (2) a noise schedule that decrease slowly enough so that for all $t$, we have $\text{dist}_\mathcal{K}(x_t) \approx \sqrt{n}\sigma_t$. For sample generation, cumulative errors introduced during the denoising process can lead to biases due to the imperfections of the denoiser (Ning et al., 2023). These errors become particularly significant by the final steps, where deviations

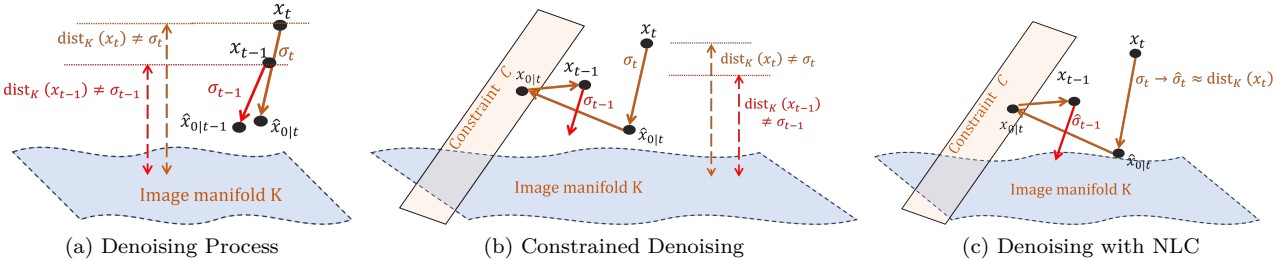

Figure 2: (a) Standard denoising using DDIM. (b) Constrained denoising interpreted as an alternative projection. In both (a) and (b), the estimated noise level $\sigma_t$ may not align with the distance $\text{dist}_{\mathcal{K}}(x_t)$. (c) Constrained denoising with noise level correction (NLC), where replacing $\sigma_t$ with a more accurate estimate $\hat{\sigma}_t$ yields projections that better align with the manifold $\mathcal{K}$.

in the denoising process result in a large mismatch between the true distance $\text{dist}_{\mathcal{K}}(x_t)$ and the estimated noise level $\sqrt{n}\sigma_t$, as illustrated in Figure 2a. As observed, a mismatch between the true distance and the estimated noise level can lead to the clean image estimation using eq. (5) falling outside the manifold $\mathcal{K}$. In constrained sample generation tasks, such as image restoration, this issue becomes more pronounced. As illustrated in Figure 2b, deviations introduced by guidance terms or projection steps onto the constraint $\mathcal{C}$ can amplify the mismatch between the estimated noise level $\sqrt{n}\sigma_t$ and the actual distance $\text{dist}_{\mathcal{K}}(x_t)$. This discrepancy is especially significant in the later denoising stages and can hinder the reconstructed sample from accurately aligning with the data manifold $\mathcal{K}$. Additional details are provided in Appendix A.

## 3.2  Noise Level Correction

To address the issue of inaccurate distance estimation caused by relying on a predefined noise level $\sigma_t$, as discussed in section 3.1, we propose a method called **noise level correction** to better align the corrected noise level with the true distance. This approach replaces the predefined noise level scheduler $\sigma_t$ with a corrected noise level $\hat{\sigma}_t$ during the denoising process, enabling more accurate distance estimation and improved sample quality. As shown in Figure 2c, using the corrected noise level brings the estimated clean sample $\hat{x}_{0|t}$ closer to the data manifold $\mathcal{K}$ compared to the naive denoising process that relies solely on $\sigma_t$. The corrected noise level is defined as $\hat{\sigma}_t := \sigma_t[1 + \hat{r}_t] \approx \text{dist}_{\mathcal{K}}(x_t)/\sqrt{n}$, where $\hat{r}_t$ represents the residual and can be modeled using either a neural network or a non-parametric function. This residual alignment approach is effective because the residual $\hat{r}_t$ is stable across noise levels, making it easier to model, while the noise level itself may become unbounded during large diffusion time steps.

For $\text{dist}_{\mathcal{K}}(x_t)$, calculating the ground-truth distance to the manifold is generally not feasible. Instead, we approximate it using the distance between the noisy sample and its clean counterpart in the forward diffusion process. Specifically, given the noisy samples $x_t$ generated by the diffusion process from $x_0 \in \mathcal{K}$ in eq. (2), we estimate the distance as $\text{dist}_{\mathcal{K}}(x_t) \approx |x_t - x_0|$. This approximation is reasonable when the projection of $x_t$ onto $\mathcal{K}$ satisfies $\text{proj}_{\mathcal{K}}(x_t) = \text{proj}_{\mathcal{K}}(x_0 + \sigma_t\epsilon) \approx x_0$. As illustrated in (Permenter & Yuan, 2024), this occurs when $\sigma_t$ is small (due to the manifold hypothesis, the random noise $\epsilon$ is orthogonal to the manifold $\mathcal{K}$), and when $\sigma_t$ is much larger than the diameter of $\mathcal{K}$.

We introduce a neural network to learn $\hat{r}_t = r_\theta(x_t, \sigma_t)$ for noise level correction. To minimize computational costs, we design the noise level correction network $r_\theta(\cdot)$ to be small and efficient. It leverages the encoder module of the denoiser's UNet architecture, followed by compact layers that fully utilize the pre-trained denoiser's capabilities. As shown in Figure 3, the denoiser network uses a UNet structure to estimate the noise vector (denoising direction) based on the noisy image $x_t$ and $\sigma_t$. Meanwhile, the noise level correction network utilizes the shared encoder, followed by additional neural network blocks, to predict the residual noise level. We train the noise level correction network $r_\theta(\cdot)$ alongside a fixed, pre-trained denoiser $\epsilon_\theta(\cdot)$, ensuring coordinated improvement in denoising accuracy. In training, to further enhance the noise level correction network, we use a randomly sampled scaling factor $\lambda$ to perturb the true noise level $\sigma_t$, expanding

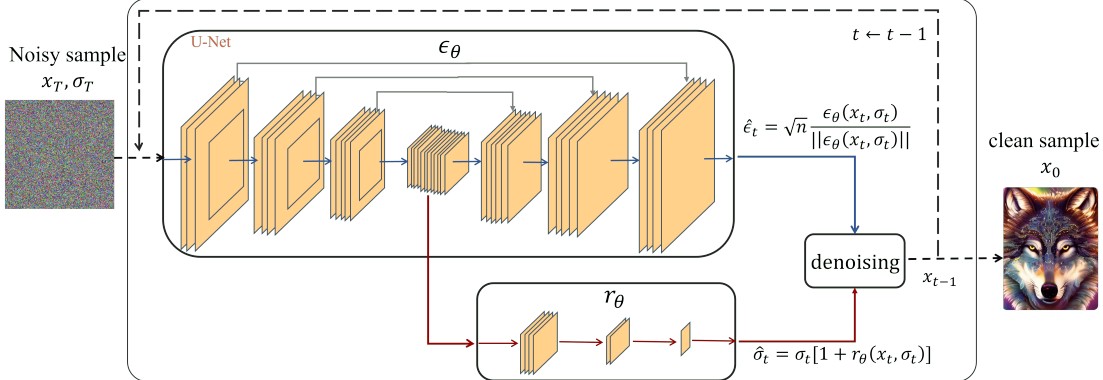

Figure 3: Architecture of the noise level correction network $r_\theta(\cdot)$ and the denoiser $\epsilon_\theta(\cdot)$.

the input-output space of $r_\theta(x_t, \sigma_t)$. The objective function for noise level correction is defined as:

$$L_{r_\theta} := \mathbf{E}_{x_0, t, \epsilon, \lambda} \left[ (\sqrt{n}\sigma_t[1 + r_\theta(\hat{x}_t, \sigma_t)] - \sigma_t\lambda\|\epsilon\|)^2 \right] \tag{14}$$

$$\hat{x}_t = x_0 + \sigma_t\lambda\epsilon, \ \epsilon \sim \mathcal{N}(0, I), \ \lambda \sim \mathcal{U}(1 - \delta, 1 + \delta) \tag{15}$$

The scaling factor $\lambda$ is sampled from a uniform distribution $\mathcal{U}(1 - \delta, 1 + \delta)$, with $\delta = 0.5$ in our experiment, to control the level of variation introduced into the noise level correction. By allowing $\lambda$ to vary, e.g., $\lambda \sim \mathcal{U}(1 - \delta, 1 + \delta)$, the true noise level becomes approximately $\lambda\sigma_t\sqrt{n}$ and $r_\theta(\hat{x}_t, \sigma_t)$ adapts to capture this variation. This approach covers a broader output space compared to the naive setting $\lambda \equiv 1$, leading to better generalization. For further discussion and an ablation study on different $\lambda$ values, please refer to appendix D.3.

### 3.3 Enhancing Sample Generation with Noise Level Correction

The trained noise level correction network $r_\theta$ can be integrated into various existing sampling algorithms to improve sample quality. For algorithms that include an initial sample estimate $\hat{x}_{0|t}$, eq. (5), we reformulate the one-step estimation as follows:

$$\hat{x}_{0|t} = x_t - \hat{\sigma}_t\hat{\epsilon}_t \tag{16}$$

$$\hat{\sigma}_t = \sigma_t[1 + r_\theta(\hat{x}_t, \sigma_t)], \ \hat{\epsilon}_t = \sqrt{n}\frac{\epsilon_\theta(x_t, \hat{\sigma}_t)}{\|\epsilon_\theta(x_t, \hat{\sigma}_t)\|} \tag{17}$$

We normalize the noise vector $\epsilon_\theta(\cdot)$ during the sampling as in eq. (17), process to decouple noise level (magnitude) correction $\hat{\sigma}_t$ from direction estimation $\hat{\epsilon}_t$. Empirical experiments show that normalizing $\epsilon_\theta(\cdot)$ and using $\hat{\sigma}_t$ to account for magnitude yields better results compared to not normalizing $\epsilon_\theta(\cdot)$. In the training loss function eq. (14), normalization of the noise vector $\epsilon$ is unnecessary because the randomly sampled noise $\epsilon$ naturally concentrates around its expected norm, approximately $\sqrt{n}$. However, the neural network-estimated noise vector $\epsilon_\theta(\cdot)$ does not necessarily maintain a constant norm.

Using eq. (17), we integrate noise level correction into the DDIM and DDPM sampling algorithms, as illustrated in Algorithm 1, with the modifications from the original DDIM/DDPM algorithms highlighted in blue. In this algorithm, lines 3 and 4 represent the current and next-step noise level corrections, respectively, while line 5 provides the normalized noise vector. Lines 6 through 8 follow the steps of the original DDIM and DDPM algorithms. Similarly, noise level correction can be integrated into the EDM sampling algorithm, as shown in Algorithm 3. Note that in EDM with noise level correction, we do not normalize the noise vector $\epsilon_\theta(x_t, \hat{\sigma}_t)$, since EDM employs a second-order Heun solver to improve noise vector estimation. By incorporating noise level correction, these algorithms produce higher-quality samples with improved accuracy by considering both the direction and distance to the data manifold during the denoising process.

---

**Algorithm 1** DDIM/DDPM with Noise Level Correction (DDIM/DDPM-NLC)

---

**Input:** Denoiser $\epsilon_\theta$ and noise level corrector $r_\theta$
**Input:** Noise scheduler $\{\sigma_t\}_{t=1}^T$, randomness scale $\eta$ ($\eta = 0$ for deterministic DDIM and $\eta = 1$ for DDPM )
**Output:** samples $x_0 \in \mathcal{K}$
1: $x_T = \sqrt{\sigma_T^2 + 1} \cdot z_T, \ z_T \sim \mathcal{N}(0, I)$,
2: **for** $t = T, T-1, \cdots, 1$ **do**
3: $\quad \hat{\sigma}_t = \sigma_t[1 + r_\theta(x_t, \sigma_t)]$
4: $\quad \hat{\sigma}_{t-1} = \hat{\sigma}_t \frac{\sigma_{t-1}}{\sigma_t}$
5: $\quad \hat{\epsilon}_t = \sqrt{n}\epsilon_\theta(x_t, \hat{\sigma}_t)/\|\epsilon_\theta(x_t, \hat{\sigma}_t)\|$
6: $\quad \sigma_{noise} = \eta\frac{\hat{\sigma}_{t-1}}{\hat{\sigma}_t}\sqrt{\hat{\sigma}_t^2 - \hat{\sigma}_{t-1}^2}$
7: $\quad \sigma_{signal} = \sqrt{\hat{\sigma}_{t-1}^2 - \sigma_{noise}^2}$
8: $\quad x_{t-1} = x_t + (\sigma_{signal} - \hat{\sigma}_t)\hat{\epsilon}_t + \sigma_{noise}\omega_t, \ \text{where} \ \omega_t \sim \mathcal{N}(0, I)$
9: **end for**

---

### 3.4 Constrained Sample Generation

The noise level correction method can also improve performance in constrained sample generation tasks, such as image restoration. Let $\mathcal{K}$ denote the data manifold, and let $\mathcal{C}$ represent specific constraints, such as masked pixel matching in inpainting tasks. Constrained sample generation aims to generate samples $x$ that satisfy both the manifold and constraint requirements, meaning $x \in \mathcal{K} \cap \mathcal{C}$. Similar to DDIM-NLC sampling approach in Algorithm 1, noise level correction can be incorporated into existing constrained sample generation methods, such as DDNM (Wang et al., 2023), a DDIM-based image restoration algorithm. An example of this is shown in Algorithm 4.

To further enhance constrained sample generation, we propose a flexible iterative projection algorithm inspired by the alternating projection technique (Bauschke & Borwein, 1996; Lewis & Malick, 2008). This approach iteratively projects samples onto each constraint set to approximate a solution that lies in the intersection of $\mathcal{K}$ and $\mathcal{C}$. The iterative projection process can be expressed as follows:

$$\hat{x}_{0|k} = \text{proj}_{\mathcal{K}}(x_{(k)}), \ x_{0|k} = \text{proj}_{\mathcal{C}}(\hat{x}_{0|k}), \ x_{(k+1)} = x_{0|k} + \bar{\epsilon}, \ , k = 0, 1, \cdots \quad (18)$$

Where $\hat{x}_{0|k}$ and $x_{0|k}$ represent the $k$-th estimates of points satisfying $x \in \mathcal{K}$ and $x \in \mathcal{C}$, respectively. The iterative rule ensures that $x_{0|k}$ approximates a point in $\mathcal{K} \cap \mathcal{C}$. $x_{k+1}$ introduces a small noise term $\bar{\epsilon}$, which helps avoid convergence to local minima in non-flat regions. This noise term is analogous to the one used in DDPM (eq. (8)) and facilitates iterative refinement toward the final clean samples. The noise term $\bar{\epsilon}$ can be gradually reduced over iterations or set to zero once a satisfactory iteration $k = K_{max}$ is achieved. At this point, the algorithm returns a final estimate such that $x_{K_{max}} \in \mathcal{K} \cap \mathcal{C}$.

Theorem 1 demonstrates that the projection operator $\text{proj}_{\mathcal{K}}(\cdot)$ can be approximately computed using the denoising diffusion models. Starting from an initial random point $x_0 = \sigma_{\max}\epsilon$, the projection onto the manifold $\mathcal{K}$ can be iteratively refined using eq. (16) and eq. (17), as follows:

$$\text{proj}_{\mathcal{K}}(x_{(k)}) = \hat{x}_{0|k} = x_{(k)} - \hat{\sigma}_{(k)}\hat{\epsilon}_{(k)} \quad (19)$$

For the additional constraint projection, $\text{proj}_{\mathcal{C}}(x)$, the specific calculation depends on the nature of the constraint. The constraints for many image restoration tasks are linear, including inpainting, colorization, super-resolution, deblurring, and compressive sensing. For tasks with linear constraints, the projection can be computed directly or optimized using gradient descent. Consider an image restoration task formulated as $y = \mathbf{A}x_0$, where $x_0$ represents the ground-truth image, $y$ is the degraded observation, $\mathbf{A}$ is the linear degradation operator. Given the degraded image $y$ and the current estimate $\hat{x}_{0|k}$, the projection onto the constraint can be computed as:

$$x_{0|k} = \text{proj}_{\mathcal{C}}(\hat{x}_{0|k}) = \mathbf{A}^\dagger y + (\mathbf{I} - \mathbf{A}^\dagger\mathbf{A})\hat{x}_{0|k} \quad (20)$$

Where $\mathbf{A}^{\dagger}$ is the pseudo-inverse of $\mathbf{A}$. In this work, we adopt the values of $\mathbf{A}^{\dagger}$ for image restoration tasks as provided in (Wang et al., 2023). Here, $x_{0|k}$ is the $k-$th th estimate satisfying $x_0 \in \mathcal{K} \cap \mathcal{C}$. This iterative estimation requires a predefined noise scheduler $\sigma_1, \cdots, \sigma_{(k)}, \cdots$ to generate $x_{0|k}$ and $x_{(k+1)}$. To allow flexible and potentially unlimited refinement steps, we define the noise schedule $\sigma_{(k)}$ with a maximum noise level $\sigma_{(0)} = \sigma_{max}$ and a minimum level $\sigma_{min}$, decaying by a predefined factor $\eta < 1$. If the noise level reaches $\sigma_{min}$, the process can either stop with returning $x_{0|k}$ or restart from $\sigma_{restart}$. This strategy permits an arbitrary number of refinement steps, stopping either at a desired loss threshold or continuing indefinitely. Since $\sigma_{(k)}$ represents the distance of noisy samples from the manifold, this decaying schedule incrementally reduces $x_{(k)}$'s distance from the manifold.

---

**Algorithm 2** Constrained Sample Generation with Noise Level Correction (IterProj-NLC)

---

**Input:** Denoiser $\epsilon_\theta$ and noise level corrector $r_\theta$
**Input:** Constraint $\mathcal{C}$, distance decay $\alpha$, noise scale $\eta$
**Input:** $\sigma_{max}, \sigma_{min}$, and $\sigma_{restart}$, and maximum iterations $K_{max}$
**Output:** samples $x_{0|K} \in \mathcal{K} \cap \mathcal{C}$
1:   $x_{(0)} = \sigma_{(0)}\epsilon, \ \epsilon \sim \mathcal{N}(0, I), \ \sigma_{(0)} = \sigma_{max}$
2:   **for** $\ k = 0, 1, 2, \cdots,$ **do**
3:      $\hat{\sigma}_{(k)} = \sigma_{(k)}[1 + r_\theta(x_{(k)}, \sigma_{(k)})]$
4:      $\hat{\epsilon}_{(k)} = \sqrt{n}\epsilon_\theta(x_{(k)}, \hat{\sigma}_{(k)})/\|\epsilon_\theta(x_{(k)}, \hat{\sigma}_{(k)})\|$
5:      $\hat{x}_{0|k} = x_{(k)} - \hat{\sigma}_{(k)}\hat{\epsilon}_{(k)}$
6:      $x_{0|k} = \text{Proj}_{\mathcal{C}}(\hat{x}_{0|k})$    (For image restoration tasks, refer eq. (20))
7:      $\sigma_{(k+1)} = \alpha\sigma_{(k)}$
8:      **if** $\sigma_{(k+1)} < \sigma_{min}$ **then**
9:        $\sigma_{(k+1)} = \sigma_{restart}$
10:     **end**
11:     $\tilde{\epsilon}_{(k)} = \sqrt{1 - \eta^2}\hat{\epsilon}_{(k)} + \eta\epsilon, \ \text{where} \ \epsilon \sim \mathcal{N}(0, I)$
12:     $x_{(k+1)} = x_{0|k} + \sigma_{(k+1)}\tilde{\epsilon}_{(k)}$
13:     **if** $k \geq K_{max}$ **or** $\|x_{0|k} - x_{0|k-1}\|$ is small enough **then**
14:       **return** $x_{0|k}$
15:     **end**
16: **end for**

---

Algorithm 2 presents the proposed constrained generation approach, termed IterProj-NLC (Iterative Projection with Noise Level Correction). Lines 3 to 5 project onto the data manifold $\mathcal{K}$, while Line 6 projects onto the constraint set $\mathcal{C}$. Lines 7 to 10 update the noise level, and Lines 11 and 12 compute the next noisy sample $x_{(k+1)}$ from the current clean estimate $x_{k|0}$. This step also acts as a convex combination of the current clean estimate $x_{k|0}$ and the previous noisy sample $x_{(k)}$.

## 4 Experiments

### 4.1 Toy Experiments

We conducted a toy experiment to demonstrate the effectiveness of noise level correction in sample generation for diffusion models. The goal was to generate samples on $d$-sphere manifold. The toy training dataset was sampled from $d$-dimensional sphere manifold embedded within an $n$-dimensional data space, where $d < n$. Detailed experimental design information is available in Appendix C.1.

After training, we applied the proposed 10-step DDIM with Noise Level Correction (DDIM-NLC), as described in Algorithm 1, and compared it against the standard 10-step DDIM baseline. Sample quality was assessed by measuring the Euclidean distance from generated samples to the ground-truth manifold $\mathcal{K}$, with lower values indicating better quality. As discussed in Theorem 3.1, the noise level $\sqrt{n}\sigma_t$ approximates the distance to the manifold. To further evaluate this approximation, we examined the alignment between the estimated noise level $\hat{\sigma}_t$ and the true distance $\text{dist}_{\mathcal{K}}(\hat{x}_t)$.

Figure 4a shows the sample-to-manifold distances over time. The dashed curve indicates the predefined noise level $\sqrt{n}\sigma_t$, while the solid curves represent the measured distances from intermediate samples $x_t$ to the manifold. In the DDIM baseline, the actual distance deviates from $\sigma_t$, especially in later steps, highlighting the inaccuracy of predefined noise level assumptions (see also Figure 2). In contrast, the DDIM-NLC consistently produces samples that are closer to the manifold, demonstrating improved alignment through noise level correction. Figure 4b quantifies this improvement using the relative distance estimation bias, defined as:

$$\text{Distance Estimation Bias} = \frac{\text{dist}_{\mathcal{K}}(\hat{x}_t) - \sqrt{n}\hat{\sigma}_t}{\sqrt{n}\sigma_t}.$$

Where $\hat{\sigma}_t = \sigma_t$ for DDIM, and $\hat{\sigma}_t = \sigma_t[1 + r_\theta(\hat{x}_t, \sigma_t)]$ for DDIM-NLC. Results show that DDIM-NLC significantly reduces estimation bias, especially in later steps when samples approach the manifold. These findings support the effectiveness of the proposed noise level correction strategy. Results for constrained sample generation can be found in Appendix C.3. The results show that DDIM-NLC achieves a significantly smaller distance estimation bias than DDIM, particularly in the later steps as samples approach the manifold. As illustrated in fig. 2, by correcting noise level estimation $\hat{\sigma}_t$, DDIM-NLC with more accurately estimates the distance. The results of constrained sample generation are shown in Appendix C.3.

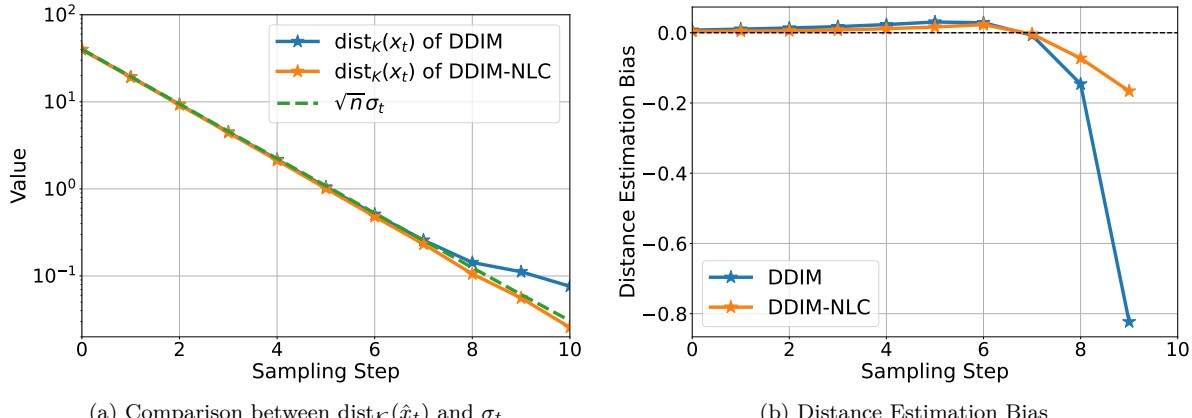

(a) Comparison between $\text{dist}_{\mathcal{K}}(\hat{x}_t)$ and $\sigma_t$      (b) Distance Estimation Bias

Figure 4: Results from the toy experiment. (a) Sample-to-manifold distance over time for DDIM and DDIM-NLC. The dashed line shows the predefined noise level $\sqrt{n}\sigma_t$. (b) Relative bias in distance estimation.

## 4.2 Unconstrained Image Generation

We conducted experiments to evaluate the effectiveness of noise level correction in unconstrained image generation tasks. The noise level correction network was trained on top of a pre-trained denoiser network. Notably, the noise level correction network is approximately ten times smaller than the denoiser network. Additional details on the experimental setup are provided in Appendix D.1. We used the FID (Fréchet Inception Distance) score as the evaluation metric to assess the quality of generated samples, where lower scores indicate better quality, following standard practice in image generation tasks (Heusel et al., 2017). The experimental results for the DDIM/DDPM framework on the CIFAR-10 dataset are shown in Table 1. As observed, DDPM-NLC and DDIM-NLC, which incorporate noise level correction, outperform the original DDPM and DDIM models across all sampling steps. Specifically, our proposed noise level correction approach improves DDIM performance by 32%, 31%, 22%, 33%, and 28% for 10, 20, 50, 100, and 300 sampling steps, respectively.

We evaluate the effectiveness of noise level correction with EDM (Karras et al., 2022), as it achieves state-of-the-art sampling quality with few sampling steps. Specifically, we assess the impact of NLC using both a first-order Euler ODE solver and a second-order Heun ODE solver within the EDM framework. As shown in Table 2, the proposed NLC also enhances the performance of EDM-based sampling methods. Notably, as a robust sampling technique, noise level correction improves the performance of the Heun sampler by 10% with just 13 sampling steps.

Table 1: FID on DDIM/DDPM sampling on CIFAR-10 with and without noise level correction.

| Method\Step | 1000 | 300 | 100 | 50 | 20 | 10 |
|---|---|---|---|---|---|---|
| DDPM | 2.99 | 2.95 | 3.37 | 4.43 | 10.41 | 23.19 |
| DDPM-NLC | **2.35** | **2.21** | **2.39** | **2.74** | 6.44 | 19.27 |
| DDIM | 4.29 | 4.32 | 4.66 | 5.17 | 8.25 | 14.21 |
| DDIM-NLC | 3.11 | 3.11 | 3.12 | 4.04 | **5.66** | **9.61** |

Table 2: FID on EDM sampling on CIFAR-10 with and w/o NLC.

| Method\Step | 35 | 21 | 13 |
|---|---|---|---|
| Euler | 3.81 | 6.29 | 12.28 |
| Euler-NLC | 2.79 | 4.21 | 8.17 |
| Heun | 1.98 | 2.33 | 7.22 |
| Heun-NLC | **1.95** | **2.22** | **6.56** |

## 4.3 Image Restoration

In this section, we evaluate the effectiveness of noise level correction on five common image restoration tasks, $4\times$ super-resolution (SR) using bicubic downsampling, deblurring with a Gaussian blur kernel, colorization using an average grayscale operator, compressed sensing (CS) with a Walsh-Hadamard sampling matrix at a 0.25 compression ratio, and inpainting with text masks. These experiments are conducted on the ImageNet (Deng et al., 2009) and CelebA-HQ (Karras et al., 2018) datasets. We compare our method with recent state-of-the-art diffusion-based image restoration methods, including ILVR (Choi et al., 2021), RePaint (Lugmayr et al., 2022), DDRM (Kawar et al., 2022), and DDNM (Wang et al., 2023). For a fair comparison, all diffusion-based methods utilize the same pretrained denoising networks with the same 100-step denoising process (100 number of inference steps), following the experimental setup in (Wang et al., 2023). To evaluate sample quality, we use FID, PSNR (Peak Signal-to-Noise Ratio), and SSIM (Structural Similarity Index Measure). For colorization, where PSNR and SSIM are less effective metrics (Wang et al., 2023), we additionally use a Consistency metric, denoted as "Cons" and calculated as $\|Ax_0 - y\|_1$. As a baseline, we also include the inverse solution for each image restoration task, given by $\hat{x} = \mathbf{A}^\dagger y$, which achieves zero constraint violation but lacks the data manifold information.

The results on the ImageNet dataset are summarized in Table 3, while those for CelebA-HQ are shown in Table 4. Tasks not supported by certain methods are marked as "N/A." As the results indicate, integrating noise level correction (as in Algorithm 4) enhances sample generation performance for DDNM. Furthermore, the proposed IterProj-NLC method (Algorithm 2), achieves the best performance across all benchmarks. For instance, IterProj-NLC outperforms the baseline DDNM in FID score by 6%, 59%, 14%, and 50% on $4\times$ SR, Deblurring, CS 25%, and Inpainting tasks, respectively. It also improves Consistency in colorization by 9%. Qualitative comparisons are shown in Figure 1, with additional results in Appendix E.1.

Table 3: Comparative results of five image restoration tasks on ImageNet.

| ImageNet Method | 4 x SR PSNR↑/SSIM↑/FID↓ | Deblurring PSNR↑/SSIM↑/FID↓ | Colorization Cons↓/FID↓ | CS 25% PSNR↑/SSIM↑/FID↓ | Inpainting PSNR↑/SSIM↑/FID↓ |
|---|---|---|---|---|---|
| $\mathbf{A}^\dagger y$ | 24.26 / 0.684 / 134.4 | 18.46 / 0.6616 / 55.42 | 0.0 / 43.37 | 15.65 / 0.510 / 277.4 | 14.52/ 0.799 / 72.71 |
| ILVR | 27.40 / 0.870 / 43.66 | N/A | N/A | N/A | N/A |
| RePaint | N/A | N/A | N/A | N/A | 31.87 / 0.968 / 13.43 |
| DDRM | 27.38 / 0.869 / 43.15 | 43.01 / 0.992 / 1.48 | 260.4 / 36.56 | 19.95 / 0.704 / 97.99 | 31.73 / 0.966 / 10.82 |
| DDNM | 27.45 / 0.870/ 39.56 | 44.93 / 0.993 / 1.17 | 42.32 / 36.32 | 21.62 / 0.748 / 64.68 | 31.60 / 0.946 / 9.79 |
| DDNM-NLC | 27.50 / 0.872 / 37.82 | 46.20 / 0.995 / 0.79 | 41.60 / 35.89 | 21.27 / 0.769 / 58.96 | 32.51 / 0.957 / 7.20 |
| IterProj-NLC | **27.56 / 0.873 / 37.48** | **48.24 / 0.997 / 0.48** | **38.30/ 35.66** | **22.27 / 0.771 / 55.69** | **33.58 / 0.966 / 4.90** |

Table 4: Comparative results of five image restoration tasks on Celeba-HQ.

| Celeba-HQ Method | 4 x SR PSNR↑/SSIM↑/FID↓ | Deblurring PSNR↑/SSIM↑/FID↓ | Colorization Cons↓/FID↓ | CS 25% PSNR↑/SSIM↑/FID↓ | Inpainting PSNR↑/SSIM↑/FID↓ |
|---|---|---|---|---|---|
| $\mathbf{A}^\dagger y$ | 27.27 / 0.782 / 103.3 | 18.85 / 0.741 / 54.31 | 0.0 / 68.81 | 15.09 / 0.583 / 377.7 | 15.57 / 0.809 / 181.56 |
| ILVR | 31.59 / 0.945 / 29.82 | N/A | N/A | N/A | N/A |
| RePaint | N/A | N/A | N/A | N/A | 35.20 / 0.981 /18.21 |
| DDRM | 31.63 / 0.945 / 31.04 | 43.07 / 0.993 / 6.24 | 455.9 / 31.26 | 24.86 / 0.876 / 46.77 | 34.79 / 0.978 /16.35 |
| DDNM | 31.63 / 0.945 / 22.50 | 46.72 / 0.996 / 1.42 | 26.25 / 26.78 | 27.52 / 0.909 / 28.80 | 35.64 / 0.979 / 12.21 |
| DDNM-NLC | 31.78 / 0.947 / 22.10 | 46.78 / 0.997 / 1.36 | 24.92 / 25.81 | 27.63 / 0.914 / 24.72 | 36.48 / 0.980 / 11.60 |
| IterProj-NLC | **31.93 / 0.949 / 21.96** | **46.97 / 0.997 / 1.29** | **24.65 / 25.30** | **27.78 / 0.916 / 23.45** | **36.57 / 0.981 / 11.07** |

### 4.4 Lookup Table for Noise Level Correction

In this section, we explore the statistical properties of the noise level correction network and demonstrate how these properties can be leveraged to create a lookup table for correcting noise levels without neural network inference. The lookup table for noise level correction is defined as $\hat{\sigma}_t = \sigma_t[1 + \hat{r}_t]$ where $\hat{r}_t$ is a non-parametric function that approximates the actual distance to the data manifold. As illustrated in the toy experiment shown in Figure 4b, distance estimation error using noise levels is lower in the initial sampling steps and increases in later stages when the true distance to the manifold decreases. This trend is expected: in the early stages, noisy samples are farther from the manifold, making approximate projections easier and reducing relative distance estimation error. More specifically, at the initial steps, the true distance $\mathrm{dist}_{\mathcal{K}}(x_t)$ is slightly larger than the estimation from the noise level $\sqrt{n}\sigma_t$ as supported by eq. (22). In later steps, however, $\mathrm{dist}_{\mathcal{K}}(x_t)$ decreases more rapidly than $\sqrt{n}\sigma_t$.

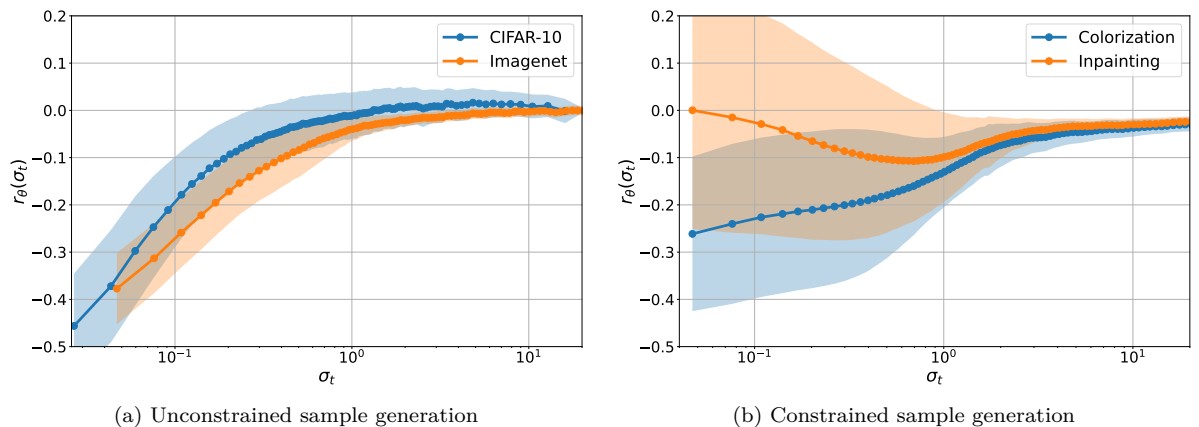

(a) Unconstrained sample generation          (b) Constrained sample generation

Figure 5: Plot of $r_\theta(\sigma_t)$ versus $\sigma_t$ in the unconstrained DDIM-NLC denoising process and constrained DDNM-NLC denoising process. The curve represents the average over samples, with shaded regions indicating the standard deviation. The larger variance (right) illustrates that the corrections applied by $r_\theta(\sigma_t)$ are too complex for a simple look-up table in the context of constrained generation.

Table 5: FID on DDIM sampling on CIFAR-10 with lookup table noise level correction.

| Method\Step | 1000 | 300 | 100 | 50 | 20 | 10 |
|---|---|---|---|---|---|---|
| DDIM | 4.29 | 4.32 | 4.66 | 5.17 | 8.25 | 14.21 |
| DDIM-LT-NLC | 4.01 | 3.97 | 3.83 | 4.37 | 6.54 | 11.21 |
| DDIM-NLC | 3.11 | 3.11 | 3.12 | 4.04 | 5.66 | 9.61 |

Table 6: FID on EDM sampling on CIFAR-10 with and with LT-NLC.

| Method\Step | 35 | 21 | 13 |
|---|---|---|---|
| Heun | 1.98 | 2.33 | 7.22 |
| Heun-LT-NLC | 1.97 | 2.27 | 6.84 |
| Heun-NLC | 1.95 | 2.22 | 6.56 |

We conducted an experiment to analyze the statistical behavior of the neural network-based noise level corrector $r_\theta(\cdot)$ for unconstrained sample generation on the CIFAR-10 and ImageNet datasets. Figure 5a presents the relationship between $r_\theta(\sigma_t)$ and $\sigma_t$, averaged over the samples $x_t$ during the DDIM-NLC denoising process. As seen, $r_\theta(\sigma_t)$ values are negative for smaller $\sigma_t$ corresponding to the final denoising steps (higher time steps $t$), and increase as $\sigma_t$ increases. This trend aligns with the observation in the toy experiment Figure 4b, indicating that distance decreases in the final steps and thus requires reducing $\hat{\sigma}_t$ for accurate distance representation. Moreover, a similar trend is observed across different datasets, such as CIFAR-10 and ImageNet. We further analyzed the statistical behavior of $r_\theta(\cdot)$ in constrained sample generation tasks on the ImageNet dataset. These tasks introduce additional variability due to constraint projections eq. (20), resulting in higher variance in $r_\theta(\cdot)$ across samples. Notably, even within the same dataset, constraints such as colorization and inpainting exhibit distinct trends during the final denoising steps (i.e., at small noise levels). Moreover, the variance at small noise levels is substantially higher in constrained tasks compared to unconstrained scenarios.

Using the values of $r_\theta(\sigma_t)$ recorded in the average value curve of Figure 5, we created a lookup table-based noise level correction (LT-NLC) search $\hat{r}_t$ to estimate $r_\theta(\sigma_t)$. We evaluated the effectiveness of LT-NLC in unconstrained sample generation tasks. The experimental results for LT-NLC applied to the DDIM framework on the CIFAR-10 dataset are shown in Table 5. As expected, the trained noise level correction (NLC) achieves the best performance. However, LT-NLC also significantly improves the original DDIM, enhancing performance by 14%, 20%, and 15% for 10, 20, and 50 sampling steps, respectively. The results for LT-NLC applied to the EDM framework on the CIFAR-10 dataset are presented in Table 6. Similar to the DDIM results, LT-NLC improves the performance of EDM-based sampling methods, demonstrating its effectiveness as a network inference-free enhancement. The results for constrained generation can be found in Appendix D.4. As illustrated in Figure 5, the variance of $r_\theta(\sigma_t)$ in constrained generation tasks, such as image restoration, is significantly higher. Consequently, the performance improvements achieved by LT-NLC are smaller compared to those of the neural network-based NLC, as LT-NLC applies the same correction across all samples. Therefore, in constrained image generation tasks, the neural network-based NLC remains essential for achieving optimal performance.

The noise level correction (NLC) network is significantly smaller than the denoiser, resulting in minimal additional computational overhead. Detailed comparisons of the training and inference times for the proposed NLC method are provided in Appendix D.2.

## 5 Conclusions

In this work, we explore the relationship between noise levels in diffusion models and the distance of noisy samples from the underlying data manifold. Building on this insight, we propose a novel noise level correction method, utilizing a neural network to align the corrected noise level with the true distance of noisy samples to the data manifold. This alignment significantly improves sample generation quality. We further extend this approach to constrained sample generation tasks, such as image restoration, within an alternating projection framework. Extensive experiments on both unconstrained and constrained image generation tasks validate the effectiveness of the proposed noise level correction network. Additionally, we introduce a lookup table-based approximation for noise level correction. This parameter-free method effectively enhances performance in various unconstrained sample generation tasks, offering a computationally efficient alternative to the neural network-based approach.

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

# A Insights for the Noise Level Correction

## A.1 Discrepancy between $\sqrt{n}\sigma_t$ and the Distance $\text{dist}_{\mathcal{K}}(x_t)$

Previous works (Ho et al., 2020; Karras et al., 2022) have primarily focused on improving the estimation of the noise vector $\epsilon_\theta(x_t, \sigma_t) \approx \epsilon_t$ eq. (4), aligning with the first assumption of a zero-error denoiser. However, these approaches typically rely on a predefined noise level scheduler $\sigma_t$ for distance estimation, often without explicit validation. This reliance can lead to inaccuracies, even at the initial step, where $\text{dist}_{\mathcal{K}}(x_T) \neq \sqrt{n}\sigma_T$.

Consider the DDIM as an example. The initial noisy point $x_T$ is sampled as follows:

$$x_T = \frac{z_T}{\sqrt{\alpha_t}} = \sqrt{\sigma_T^2 + 1} \cdot z_T, \ z_T \sim \mathcal{N}(0, I) \tag{21}$$

Let $x_0^* = \text{Proj}_{\mathcal{K}}(x_t) \in \mathcal{K}$ denotes the projection of $x_t$ onto the manifold $\mathcal{K}$. In the context of an image manifold, we have $\|x_0^*\| > 0$. If $\langle z_T, x_0^* \rangle \leq 0$, the expectation of squared distance is given by:

$$\begin{aligned} \mathbf{E}\left[\|x_T - x_0^*\|^2\right] = \mathbf{E}\left[\|\sqrt{\sigma_T^2 + 1} \cdot z_T - x_0^*\|^2\right] &= \mathbf{E}\left[(\sigma_T^2 + 1)\|z_T\|^2 + \|x_0^*\|^2 - 2\sqrt{\sigma_T^2 + 1}\langle z_T, x_0^*\rangle\right] \\ &\geq (\sigma_T^2 + 1)\mathbf{E}[\|z_T\|^2] + \mathbf{E}[\|x_0^*\|^2] \\ &= (\sigma_T^2 + 1)n + \mathbf{E}[\|x_0^*\|^2] > \sigma_T^2 n \end{aligned} \tag{22}$$

where the final equality uses the fact that $\mathbf{E}[|z_T|^2] = n$ for for $z_T \sim \mathcal{N}(0, I^{n \times n})$. Equation (22) implies that, with high probability, $\text{dist}_{\mathcal{K}}(x_T) > \sqrt{n}\sigma_T$. Consequently, at any step $t = T, \ldots, 0$, these deviations may result in $\text{dist}_{\mathcal{K}}(x_t) \neq \sqrt{n}\sigma_t$, potentially causing the final sample to deviate from the manifold $\mathcal{K}$. These errors arise from imperfections in the denoiser and become particularly significant in the final steps. The toy experiments on the discrepancy between noise level and true distance to the manifold can be found in appendix C.2. It is worth noting that the EDM sampling method initializes with a random step as $x_T = \sigma_T z_T$. However, it leads to the same conclusion: for $\langle z_T, x_0^* \rangle \leq 0$, we have $\text{dist}_{\mathcal{K}}(x_T) \neq \sqrt{n}\sigma_T$.

## A.2 Training Objective

To address the issue of inaccurate distance estimation caused by relying on a predefined noise level $\sigma_t$, we propose a method called *noise level correction (NLC)* to better align the estimated noise level $\hat{\sigma}_t$ with the true distance $\text{dist}_{\mathcal{K}}(x_t)/\sqrt{n}$. Specifically, we train a neural network to minimize the following objective (initial form):

$$L := \mathbb{E}\left[\left(\sqrt{n}\hat{\sigma}_t - \text{dist}_{\mathcal{K}}(x_t)\right)^2\right] \tag{23}$$

Inspired by residual learning, instead of directly predicting the distance, we propose to learn the residual between the true distance and the predefined noise level. We define:

$$\hat{\sigma}_t := \sigma_t \left[1 + r_\theta(x_t, \sigma_t)\right] \approx \frac{\text{dist}_{\mathcal{K}}(x_t)}{\sqrt{n}}, \tag{24}$$

where $r_\theta(x_t, \sigma_t)$ is the residual correction network to be trained. This residual formulation is advantageous because $r_\theta(\cdot)$ is typically more stable across noise levels, while $\sigma_t$ may vary widely during large diffusion time steps.

Since computing the true distance $\text{dist}_{\mathcal{K}}(x_t)$ is impractical, we approximate it by the distance between a noisy sample and its clean counterpart obtained from the forward diffusion process. Given $x_t = x_0 + \sigma_t \epsilon$, with $x_0 \in \mathcal{K}$ and $\epsilon \sim \mathcal{N}(0, I)$, we approximate:

$$\text{dist}_{\mathcal{K}}(x_t) \approx \|x_t - x_0\|. \tag{25}$$

This approximation is reasonable under the manifold hypothesis, assuming that the noise vector $\epsilon$ is orthogonal to the manifold, and that $\text{proj}_{\mathcal{K}}(x_t) \approx x_0$. To further improve robustness, we introduce a perturbation scaling factor $\lambda$ and expand the input-output domain of $r_\theta(\cdot)$ via:

$$x_t = x_0 + \sigma_t \lambda \epsilon, \quad \epsilon \sim \mathcal{N}(0, I), \quad \lambda \sim \mathcal{U}(1 - \delta, 1 + \delta), \tag{26}$$

where $\delta$ controls the perturbation range (e.g., $\delta = 1$ recovers the original diffusion process). This leads to the approximation:

$$\text{dist}_{\mathcal{K}}(x_t) \approx \|x_t - x_0\| = \sigma_t \lambda \|\epsilon\|. \tag{27}$$

Combining eq. (23), eq. (24), and eq. (27), we arrive at the final training objective, as shown in eq. (14):

$$L_{r_\theta} := \mathbf{E}_{x_0, t, \epsilon, \lambda} \left[ (\sqrt{n} \sigma_t [1 + r_\theta(\hat{x}_t, \sigma_t)] - \sigma_t \lambda \|\epsilon\|)^2 \right] \tag{28}$$

$$\hat{x}_t = x_0 + \sigma_t \lambda \epsilon, \ \epsilon \sim \mathcal{N}(0, I), \ \lambda \sim \mathcal{U}(1 - \delta, 1 + \delta) \tag{29}$$

During training, we sample $x_0$ from the training set, $t$ from a uniform diffusion time range (e.g., $t \in [1, 1000]$), and draw $\epsilon \sim \mathcal{N}(0, I)$ and $\lambda \sim \mathcal{U}(1 - \delta, 1 + \delta)$. The loss is computed according to eq. (14) and used to optimize the noise level correction network $r_\theta(\cdot)$.

## B   Sampling with Noise Level Correction

The proposed noise level correction (NLC) method is broadly compatible with a wide range of existing diffusion sampling algorithms and can be seamlessly integrated to enhance sample quality. In many diffusion samplers, each denoising step consists of two substeps: (1) estimating the original clean sample $\hat{x}_{0|t}$ given a noisy observation $x_t$, and (2) adding noise to propagate the sample to the next timestep. Our NLC method fits naturally into this structure by correcting the noise level during the clean sample estimation step, as described in eqs. (16) and (17). In sections 3.3 and 3.4, we demonstrated the integration of NLC with DDIM/DDPM and constrained sampling tasks. In this section, we extend our discussion to other samplers and show how the NLC method can be incorporated. All modifications to the original algorithms are highlighted in blue.

Algorithm 3 shows how NLC is integrated into the EDM sampler (Karras et al., 2022). Unlike DDIM-NLC in algorithm 1, EDM-NLC does not normalize the noise vector $\epsilon_\theta(x_t, \hat{\sigma}_t)$, since EDM is based on a second-order ODE solver. Nevertheless, the NLC method can be applied by modifying the noise level estimation step, as shown in blue.

---

**Algorithm 3** EDM with Noise Level Correction (EDM-NLC)

---

**Input:** Denoiser $\epsilon_\theta$ and noise level corrector $r_\theta$
**Input:** Noise scheduler $\{\sigma_t\}_{t=1}^T$
**Output:** samples $x_0 \in \mathcal{K}$
 1: $x_T = \sigma_T \epsilon, \ \epsilon \sim \mathcal{N}(0, I),$
 2: **for** $t = T, T - 1, \cdots, 1$ **do**
 3:     $\hat{\sigma}_t = \sigma_t [1 + r_\theta(x_t, \sigma_t))]$
 4:     $\hat{\sigma}_{t-1} = \hat{\sigma}_t \frac{\sigma_{t-1}}{\sigma_t}$
 5:     $\hat{\epsilon}_t = \epsilon_\theta(x_t, \hat{\sigma}_t)$
 6:     $x_{t-1} = x_t + (\hat{\sigma}_{t-1} - \hat{\sigma}_t)\hat{\epsilon}_t$
 7:     **if** $t > 1$ **then**
 8:        $\hat{\epsilon}_{t-1} = \epsilon_\theta(x_{t-1}, \hat{\sigma}_{t-1})$
 9:        $\bar{\epsilon}_t = 0.5\hat{\epsilon}_t + 0.5\hat{\epsilon}_{t-1}$
10:        $x_{t-1} = x_t + (\hat{\sigma}_{t-1} - \hat{\sigma}_t)\bar{\epsilon}_t$
11:     **end**
12: **end for**

---

Algorithm 4 presents the incorporation of NLC into DDNM, a constrained image generation algorithm for linear inverse problems of the form $y = \mathbf{A}x$ (Wang et al., 2023). This demonstrates that the NLC method can also enhance constrained sampling tasks.

---

**Algorithm 4** DDNM with Noise Level Correction (DDNM-NLC)

---

**Input:** Denoiser $\epsilon_\theta$ and noise level corrector $r_\theta$
**Input:** Noise scheduler $\{\sigma_t\}_{t=1}^T$, randomness scale $\eta$,
**Input:** Linear degradation operator $\mathbf{A}$, and pseudo-inverse operator $\mathbf{A}^\dagger$ for image restoration constraint $\mathcal{C}$,
**Input:** Degraded image $y$,
**Output:** samples $x_0 \in \mathcal{K} \cap \mathcal{C}$
 1: $x_T = \sqrt{\sigma_T^2 + 1} \cdot z_T, \ z_T \sim \mathcal{N}(0, I)$,
 2: **for** $t = T, T-1, \cdots, 1$ **do**
 3:     $\hat{\sigma}_t = \sigma_t[1 + r_\theta(x_t, \sigma_t)]$
 4:     $\hat{\sigma}_{t-1} = \hat{\sigma}_t \frac{\sigma_{t-1}}{\sigma_t}$
 5:     $\hat{\epsilon}_t = \sqrt{n}\epsilon_\theta(x_t, \hat{\sigma}_t)/\|\epsilon_\theta(x_t, \hat{\sigma}_t)\|$
 6:     $\hat{x}_{0|t} = x_t - \hat{\sigma}_t\hat{\epsilon}_t$
 7:     $x_{0|t} = \mathbf{A}^\dagger y + (\mathbf{I} - \mathbf{A}^\dagger\mathbf{A})\hat{x}_{0|t}$
 8:     $\sigma_{noise} = \eta\frac{\hat{\sigma}_{t-1}}{\hat{\sigma}_t}\sqrt{\hat{\sigma}_t^2 - \hat{\sigma}_{t-1}^2}$
 9:     $\sigma_{signal} = \sqrt{\hat{\sigma}_{t-1}^2 - \sigma_{noise}^2}$
10:     $x_{t-1} = x_{0|t} + \sigma_{signal}\hat{\epsilon}_t + \sigma_{noise}\omega_t, \ \text{where } \omega_t \sim \mathcal{N}(0, I)$
11: **end for**

---

Furthermore, the NLC method can be extended to other advanced samplers, such as DPM-Solver (ODE solver for Diffusion probabilistic models) (Lu et al., 2022). In algorithm 5, we present the integration of NLC into the second-order DPM-Solver. As with EDM-NLC, the primary modification involves adjusting the noise level using the NLC module. The DPM-Solver uses noise schedules $\{\alpha_t\}_{t=1}^T$ and $\{\sigma_t\}_{t=1}^T$, and defines the auxiliary parameter $\lambda_t := \log(\alpha_t/\sigma_t)$. The inverse mapping from $\lambda$ to $t$ is denoted by $t_\lambda(\cdot)$, and we define the interval $h_t := \lambda_t - \lambda_{t-1}$. These examples collectively demonstrate the flexibility of the proposed noise level correction strategy, making it a broadly applicable enhancement across various diffusion model sampling frameworks.

---

**Algorithm 5** DPM-Solver with Noise Level Correction (DPM-NLC)

---

**Input:** Denoiser $\epsilon_\theta$ and noise level corrector $r_\theta$
**Input:** Noise scheduler $\{\alpha_t\}_{t=1}^T$ and $\{\sigma_t\}_{t=1}^T$
**Output:** samples $x_0 \in \mathcal{K}$
 1: $x_T = \sigma_T\epsilon, \ \epsilon \sim \mathcal{N}(0, I)$,
 2: **for** $t = T, T-1, \cdots, 1$ **do**
 3:     $\hat{\sigma}_t \leftarrow \sigma_t \cdot [1 + r_\theta(x_t, \sigma_t)]$
 4:     $s = t_\lambda\left(\frac{\lambda_{t+1} + \lambda_t}{2}\right)$
 5:     $\hat{\sigma}_s = \hat{\sigma}_t\frac{\sigma_s}{\sigma_t}$
 6:     $\hat{\epsilon}_t = \epsilon_\theta(x_t, \hat{\sigma}_t)$
 7:     $u = \frac{\alpha_s}{\alpha_t}x_t - \hat{\sigma}_s\left(e^{\frac{h_t}{2}} - 1\right)\hat{\epsilon}_t$
 8:     $\hat{\epsilon}_s = \epsilon_\theta(u, \hat{\sigma}_s)$
 9:     $x_{t-1} = \frac{\alpha_{t-1}}{\alpha_t}x_t - \sigma_{t-1}\left(e^{h_t} - 1\right)\hat{\epsilon}_s$
10: **end for**

---

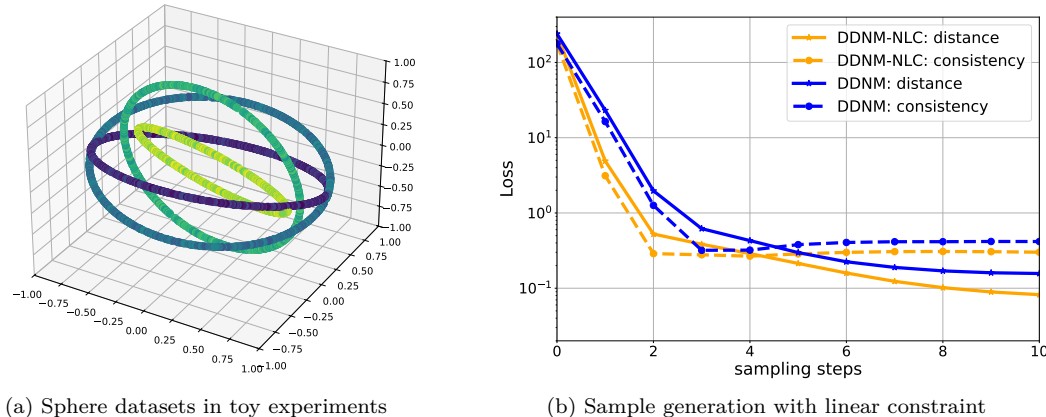

(a) Sphere datasets in toy experiments    (b) Sample generation with linear constraint

Figure 6: (a) Sphere datasets in toy experiments; (b) shows the results of sample generation with linear constraint $Ax = b$.

## C   Toy Experiments

### C.1   Experimental design for sphere manifold.

The dataset consists of samples from a $d$-dimensional sphere manifold, denoted as $s \sim \mathcal{S}^d$ embedded within an $n$-dimensional data space. To create training samples $x$ we apply $m$ different linear projections (rotations) to the original sphere $s$ and add a small amount of Gaussian noise $x_{\text{noise}}$ to the $d$-sphere signal $x_{\text{signal}}$. Let $\mathcal{K}$ represent the $d$-sphere manifold, such that $x_{\text{signal}} \in \mathcal{K}$. Each training sample $x \in \mathbb{R}^n$ is generated according to the following equations:

$$x = x_{\text{signal}} + x_{\text{noise}}, \ x_{\text{noise}} \sim \mathcal{N}(0, 10^{-6}I) \tag{30}$$

$$x_{\text{signal}} = R_k s \in \mathbb{R}^n, \text{where } k \sim \mathcal{U}(\{1, 2, \cdots, m\}), \ R_k R_k^T = I^{(n \times n)} \tag{31}$$

$$s \sim \mathcal{S}^d, \text{where } \sum_{i=1}^{d+1} s_i^2 = 1, s_{d+2} = s_{d+3} = \cdots = s_n = 0 \tag{32}$$

Where $\mathcal{N}(0, \Sigma)$ denotes the normal distribution, $\mathcal{U}(\{\cdot\})$ denotes the uniform distribution. $\mathcal{S}^d$ refers to a $d$-dimensional sphere. The matrices $R_1, R_2, \cdots, R_m$ are fixed random orthogonal matrices that are utilized to "hide" zeros in certain coordinates. For our experiments, we focus primarily on a dataset with parameters $n = 100$, $d = 1$, and $m = 4$, resulting in a 100-dimensional dataset composed of 4 circles. For illustration, we also generate a 3-dimensional dataset with parameters ($n = 3$, $d = 1$, $m = 4$), as shown in Figure 6a. A total of 10,000 samples were used to train the model. The denoiser $\epsilon_\theta(\cdot)$ is implemented with a fully connected network containing 5 layers, each with a hidden dimension of 128. For noise level correction, $r_\theta(\cdot)$ is implemented with a 2-layer fully connected network, also using a hidden dimension of 128.

### C.2   Misalignment between noise level and true distance to the manifold

As discussed in section 3.1, diffusion-based sample generation methods, such as DDIM and DDPM, often suffer from cumulative errors introduced during the denoising process. These errors arise from imperfections in the denoiser and become particularly significant in the final steps. As a result, there can be a substantial mismatch between the true distance to the data manifold, $\text{dist}_{\mathcal{K}}(x_t)$, and the estimated noise level, $\sqrt{n}\sigma_t$. To visualize this phenomenon, we conduct a toy experiment. Figure 7 shows the deviation between the true distance $\text{dist}_{\mathcal{K}}(\hat{x}_t)$ and the estimated noise level $\sqrt{n}\sigma_t$ during a 10-step DDIM sampling process. As observed, the actual distance increasingly deviates from the predefined noise level, especially in the later steps. This result aligns with the noise estimation inaccuracy illustrated earlier in fig. 2.

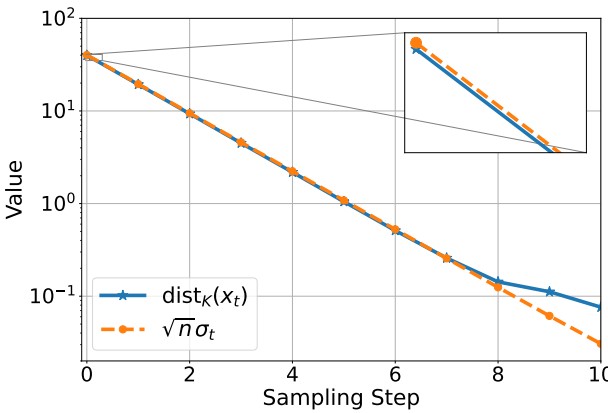

Figure 7: Deviation between the true distance $\text{dist}_{\mathcal{K}}(\hat{x}_t)$ and the estimated noise level $\sqrt{n}\sigma_t$ in DDIM sampling.

### C.3 Results of constrained sample generation.

In Section 4.1, we demonstrate the effectiveness of noise level correction in unconstrained sample generation. Here, we extend the evaluation to constrained generation, using a linear constraint $Ax = b$ with $A \in \mathbb{R}^{1 \times n}$ as a random variable and $b = 0$. We applied the proposed 10-step DDNM with Noise Level Correction (DDNM-NLC), as detailed in Algorithm 4 to generate samples and compared this method to the 10-step DDNM baseline. We report the distance (to measure sample quality) and Consistency error $\|Ax - b\|$ (to measure constraint satisfaction) as shown in Figure 6b. The proposed method shows superior performance in both metrics, generating high-quality samples that satisfy the constraint more effectively than the baseline.

## D Image Generation Experiments

### D.1 Experimental design

Table 7: Architecture of the noise level correction network $r_\theta(\cdot)$

| Network | Hyperparameters | CIFAR-10 | ImageNet | Celeba-HQ |
|---|---|---|---|---|
| Network Architecture for $r_\theta(\cdot)$ | Input Channel (feature channel) | 256 | 1024 | 512 |
| | Input Size (feature size) | 4 | 8 | 8 |
| | Residual Blocks | 2 | 2 | 2 |
| | Attention Blocks | 1 | 1 | 1 |
| | Attention Heads | 4 | 4 | 4 |
| # Parameters | $r_\theta(\cdot)$ | 14 M | 234 M | 59 M |
| | $\epsilon_\theta(\cdot)$ *(Frozen, not trained)* | 218 M | 2109 M | 434 M |

Table 8: Hyperparameters used for the training

| Settings | | CIFAR-10 | ImageNet | Celeba-HQ |
|---|---|---|---|---|
| Hyperparameters | Batch size | 128 | 64 | 64 |
| | Learning rate | 0.0003 | 0.0003 | 0.0003 |
| | # Iterations | 150 K | 300 K | 300 K |
| # Samples | Training $r_\theta(\cdot)$ | 19.2 M | 19.2 M | 19.2 M |
| | *Training $\epsilon_\theta(\cdot)$* | $\approx$ 200 M | $\approx$ 2500 M | $\approx$ 1000 M |

**Implementation.** In this experiment, we used a pretrained denoiser $\epsilon_\theta(\cdot)$, and trained the noise level correction network $r_\theta(\cdot)$ to enhance the denoising process. For unconstrained image generation experiments, we employed the pretrained $32 \times 32$ denoising network from (Song et al., 2021a) in DDIM-based experiments

on the CIFAR-10 dataset, and the pretrained $32 \times 32$ denoising network from (Karras et al., 2022) in EDM-based experiments on CIFAR-10. For image restoration experiments, we utilized the pretrained $256 \times 255$ denoising network from (Dhariwal & Nichol, 2021) for ImageNet experiments and the pretrained $256 \times 255$ denoising network from (Lugmayr et al., 2022) for CelebA-HQ experiments.

**Network architecture.** The noise correction network $r_\theta(\cdot)$ is designed to be significantly smaller than the denoiser $\epsilon_\theta(\cdot)$, while still incorporating residual and attention blocks similar to those used in the original DDPM denoiser (Ho et al., 2020). Table 7 outlines the architecture of the noise correction network for CIFAR-10, ImageNet, and CelebA-HQ datasets. For all datasets, we employ 2 residual blocks, 1 attention block, and 4 attention heads. The primary architectural difference lies in the varying feature sizes (input channels and input dimensions) generated by the denoiser's encoder. For comparison, the ImageNet denoiser contains 9 attention blocks in the encoder and 13 in the decoder. Consequently, the noise correction network is approximately ten times smaller than the denoiser network. The last two rows of Table 7 provide a parameter count comparison between the noise correction network $r_\theta$ and the denoiser $\sigma_\theta$.

**Training.** The noise correction network is trained following the original DDPM training procedure. Key hyperparameters for training are listed in Table 8. Notably, the noise correction network is trained until 19.2 million samples have been drawn from the training set. In comparison, training the denoiser requires 200 million samples for CIFAR-10 and over 2000 million samples for ImageNet.

## D.2 Time and Memory Cost

Table 9: Inference Time Overhead of the Proposed NLC Method.

| $\epsilon$ model | CIFAR10 - DDPM | CIFAR10-EDM | CelebA - DDNM | ImageNet-DDNM |
|---|---|---|---|---|
| Baseline Inference (s) | 0.035 | 0.022 | 0.049 | 0.102 |
| +NLC (s) | 0.037 | 0.023 | 0.050 | 0.104 |
| Time Overhead Ratio(%) | 5.7 | 4.3 | 2.0 | 2.0 |

Table 10: Inference Time Comparison Between the Baseline and the Proposed NLC Method. For the CIFAR-10 experiment, the baseline is DDPM and the proposed method is DDPM+NLC. For the ImageNet experiment, the baseline is DDNM and the proposed method is DDNM+NLC (inpainting experiments).

| $\epsilon$ model | CIFAR10-DDPM (Song et al., 2021a) | ImageNet - DDNM (Dhariwal & Nichol, 2021) |
|---|---|---|
| baseline | 0.32 | 0.93 |
| baseline + NLC | 0.34 | 0.95 |

**Training time**. The training time is shown in the last two rows of Table 8. This results in significantly faster training for the noise level correction network. For example, training the ImageNet denoiser on 8 Tesla V100 GPUs takes approximately two weeks, while training the noise correction network requires only about one day. This efficiency is due to the smaller size of the noise level correction network and its use of the pretrained denoiser.

**Inference time**. The proposed noise level correction (NLC) method requires only one additional inference step for the NLC network $r_\theta(\cdot)$ per overall inference. This extra step is independent of the specific sampling algorithm, applicable to both unconstrained and constrained approaches, as shown in line 3 of algorithms 1, 2, 3 and 4. Table 9 summarizes the additional time cost in different settings, indicating that each inference with the proposed NLC incurs an overhead of less than 6%. Furthermore, Table 10 presents the inference times for a real 10-step sample generation with a batch size of 1 for CIFAR-10 unconstrained sample generation with DDPM and ImageNet constrained (inpainting) generation with DDNM, confirming that incorporating the noise level correction results in a modest increase of approximately 6% in inference time.

Table 11: FID score on the scaling factor $\lambda$ in training $r_\theta(\hat{x}_t, \sigma_t)$. Here, $\delta = 0$ corresponds to the naive setting ($\lambda = 1$). CIFAR-10 experiments use 100-step sampling with DDIM-NLC, and ImageNet inpainting experiments use 100-step sampling with DDNM-NLC.

| $\delta$ | 0 | 0.2 | **0.5** | 1.0 |
|---|---|---|---|---|
| CIFAR-10 (DDIM-NLC) | 3.52 | 3.23 | **3.12** | 3.14 |
| ImageNet Inpainting (DDNM-NLC) | 7.87 | 7.59 | **7.20** | 7.24 |

### D.3  Ablation Study on the Scaling Factor $\lambda$

In the training objective for the NLC network $r_\theta(\cdot)$, we introduce the scaling factor $\lambda$ to adjust the input-output range. Specifically, we use the predicted noise level $\sqrt{n}\sigma_t[1 + r_\theta(\hat{x}_t, \sigma_t)]$ to estimate the true noise level (i.e., the distance from the noisy sample to the clean sample) $\|x_t - x_0\|$. This adjustment addresses the mismatch between the noise scales in the forward (diffusion) and reverse (denoising) processes. Noisy samples are generated as $\hat{x}_t = x_0 + \sigma_t \lambda \epsilon_t$, where $\epsilon_t \sim \mathcal{N}(0, I)$ and typically $\|\epsilon_t\| \approx \sqrt{n}$. Without scaling (i.e., setting $\lambda \equiv 1$), the true noise level becomes concentrated around $\sigma_t \sqrt{n}$, leading to $r_\theta(\hat{x}_t, \sigma_t) \approx 0$. However, the denoiser network $\epsilon_\theta(x_t, \sigma_t)$ does not necessarily predict a noise level with such a concentrated norm. This norm mismatch, also noted in (Permenter & Yuan, 2024), motivates our introduction of $\lambda$.

By allowing $\lambda$ to vary, e.g., $\lambda \sim \mathcal{U}(1 - \delta, 1 + \delta)$, the true noise level becomes approximately $\lambda \sigma_t \sqrt{n}$ and $r_\theta(\hat{x}_t, \sigma_t)$ reflects the variation in $\lambda$ (i.e., roughly in $[-\delta, \delta]$). To validate this design, we conduct an ablation study comparing different values of $\delta$ using 100-step sampling in terms of FID score, as shown in table 11. For CIFAR-10, the DDIM-NLC method is used for unconstrained sampling, while for ImageNet inpainting experiments the DDNM-NLC method is employed. As can be seen, $\delta = 0.5$ (the standard setting of our experiment) outperforms the naive setting of $\delta = 0$.

### D.4  Lookup table method for image restoration

Table 12: Comparative results of image restoration tasks on ImageNet for lookup-table noise level correction.

| ImageNet Method | 4 x SR PSNR↑/SSIM↑/FID↓ | Deblurring PSNR↑/SSIM↑/FID↓ | Colorization Cons↓/FID↓ | CS 25% PSNR↑/SSIM↑/FID↓ | Inpainting PSNR↑/SSIM↑/FID↓ |
|---|---|---|---|---|---|
| DDNM | 27.45 / 0.870/ 39.56 | 44.93 / 0.993 / 1.17 | 42.32 / 36.32 | 21.62 / 0.748 / 64.68 | 31.60 / 0.946 / 9.79 |
| DDNM-LT-NLC | 27.47 / 0.870/ 39.03 | 45.14 / 0.993 / 1.02 | 42.12 / 36.07 | 21.48 / 0.751 / 62.64 | 31.84 / 0.951 / 9.19 |
| **DDNM-NLC** | 27.50 / 0.872 / 37.82 | 46.20 / 0.995 / 0.79 | 41.60 / 35.89 | 21.27 / 0.769 / 58.96 | 32.51 / 0.957 / 7.20 |

We evaluate the performance of the lookup table noise level correction (LT-NLC) in image restoration tasks. The results on the ImageNet dataset are summarized in table 12. As shown, the DDNM image restoration method achieves additional performance gains with LT-NLC. However, these improvements are notably smaller compared to those achieved with the neural network-based NLC method.

### D.5  Additional experiments on integrating NLC with DPM

Table 13: FID scores on CIFAR-10 using DPM sampling with and without NLC, across varying numbers of function evaluations (NFE).

| Method\NFE | 12 | 18 | 24 | 30 | 36 |
|---|---|---|---|---|---|
| DPM | 5.28 | 3.43 | 3.02 | 2.85 | 2.78 |
| DPM-NLC | **4.83** | **3.22** | **2.97** | **2.81** | **2.77** |

As described in algorithm 5, we integrate the proposed Noise Level Correction (NLC) method into the widely used DPM-Solver sampling algorithm. To evaluate the effectiveness of this integration, we conduct experiments on the CIFAR-10 image generation task. Table 13 presents the FID scores for DPM with

and without NLC under different numbers of function evaluations (NFE). As shown, incorporating NLC consistently improves sample quality across all settings. For instance, with NFE = 12, NLC reduces the FID score from 5.28 to 4.83, corresponding to a 9% improvement. These results demonstrate that NLC can effectively enhance the performance of DPM-based sampling methods.

# E   Qualitative study

## E.1   Image Restoration

We present qualitative comparisons between the proposed method, IterProj-NLC, and the baseline, DDNM, across various image restoration tasks. These tasks include compressive sensing, shown in fig. 8, colorization, shown in fig. 9, inpainting, shown in fig. 10, and super-resolution, shown in fig. 11. The comparisons demonstrate the effectiveness of IterProj-NLC in producing visually superior results over the baseline.

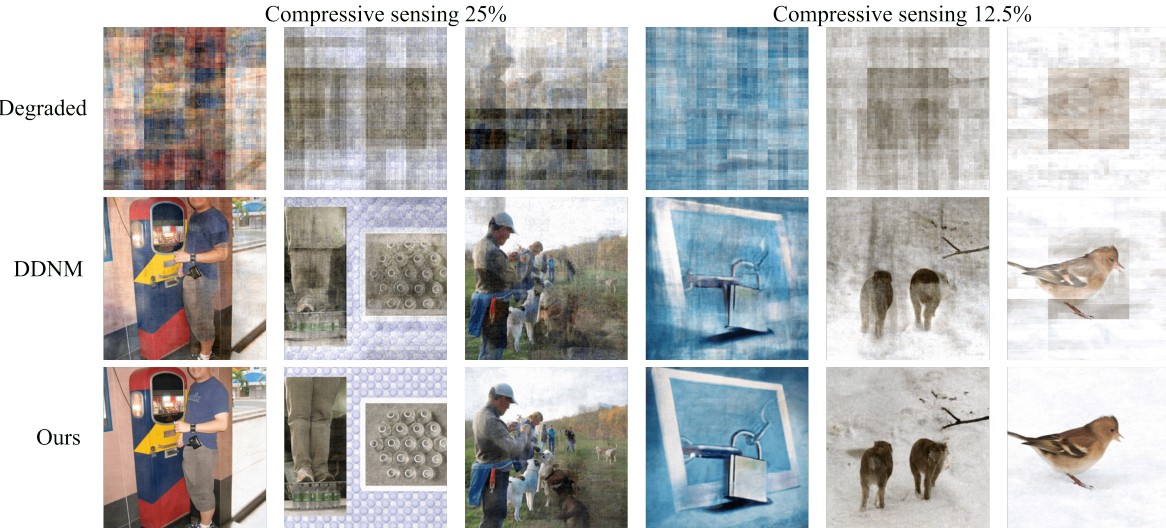

Figure 8: Qualitative results of compressive sensing.

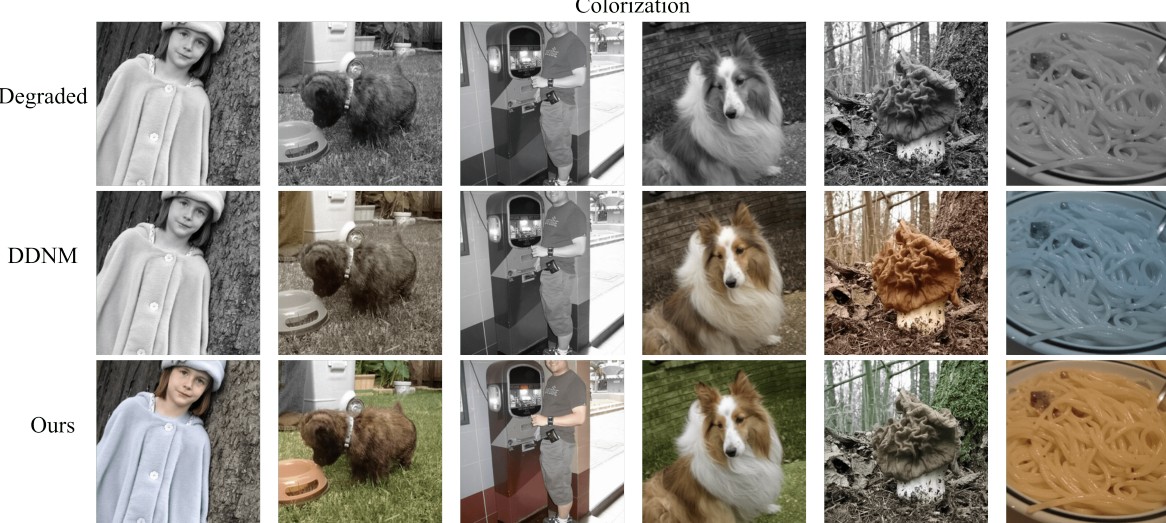

Figure 9: Qualitative results of colorization.

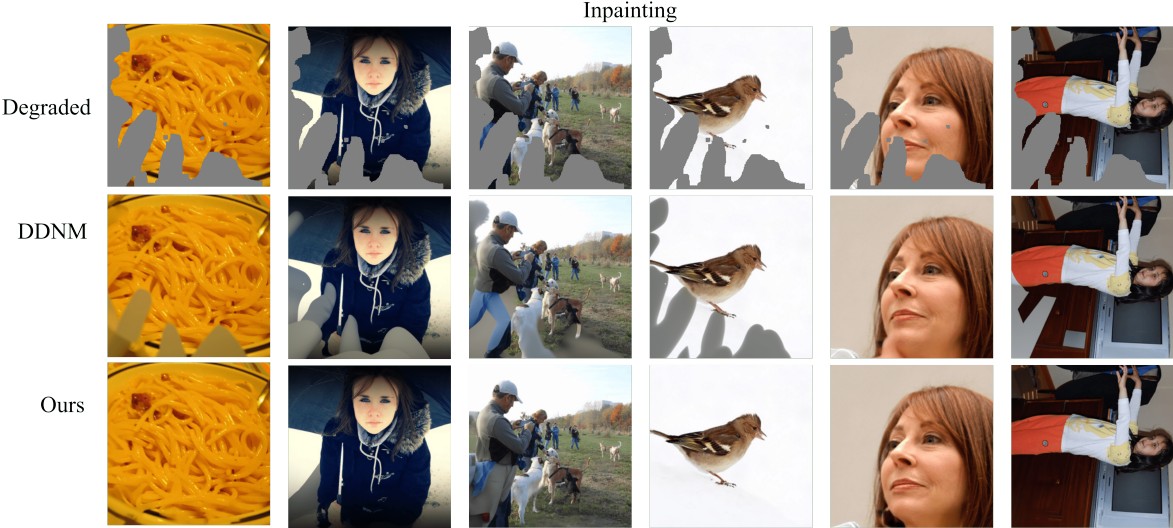

Figure 10: Qualitative results of inpainting.

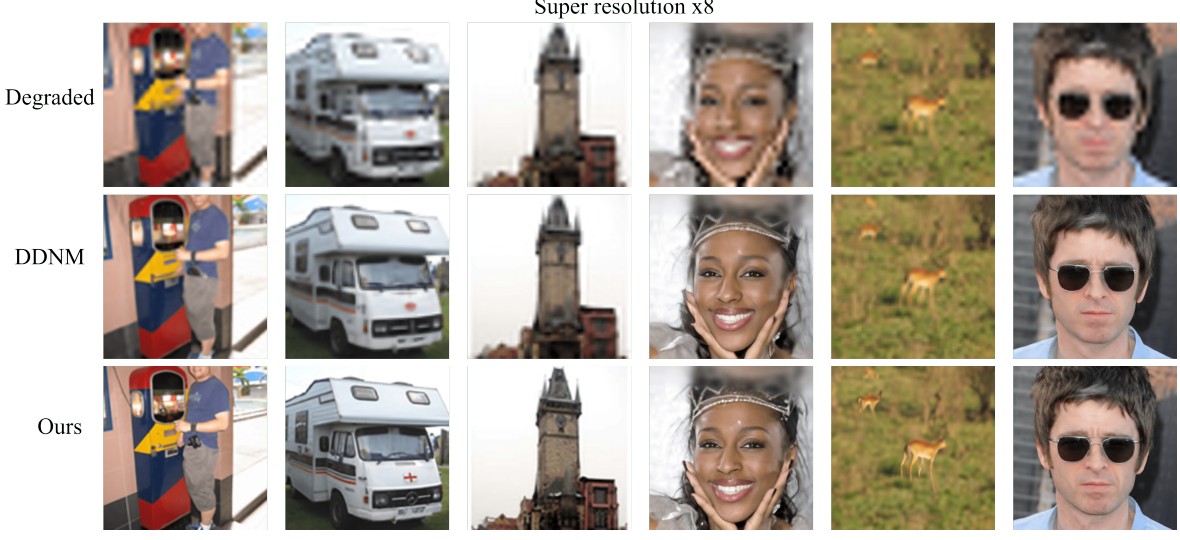

Figure 11: Qualitative results of super resolution.

### E.2 Unconstrained Image Generation

The example results of CIFAR-10 generated using 100-step sampling of DDIM-NLC are presented in fig. 12.

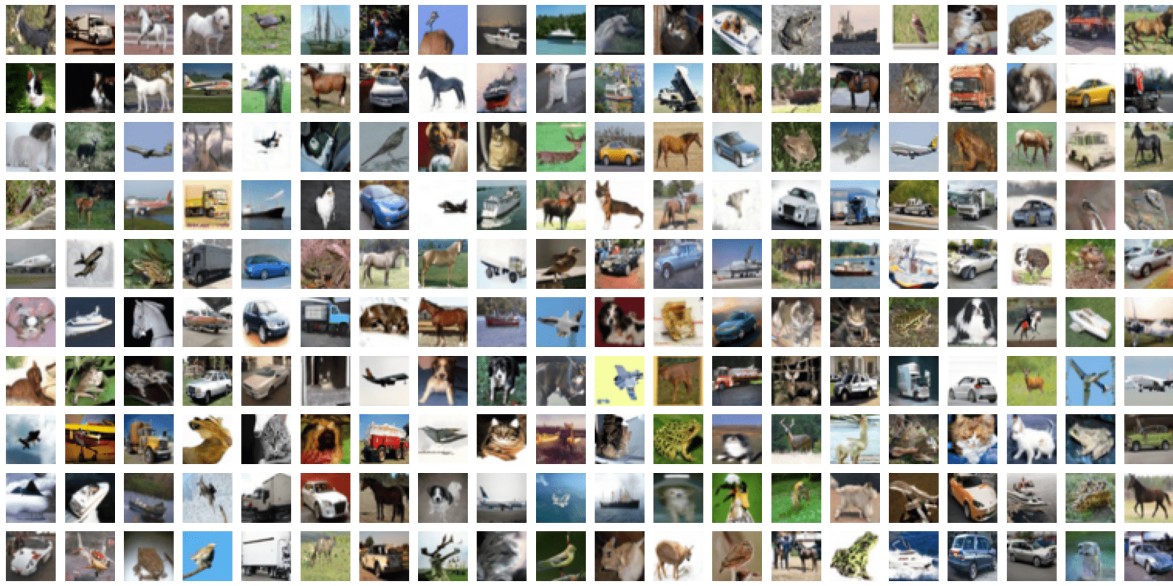

Figure 12: Example results of CIFAR-10 generated using DDIM-NLC.

