# OpenReview forum: "Enhancing Sample Generation of Diffusion Models using Noise Level Correction"
_TMLR — Accepted by TMLR_

### Review · Reviewer_A2wi · 2025-02-19

**Summary Of Contributions:**

The authors aim to improve the quality of generated samples by diffusion models.

For this, they train a noise level correction (NLC) network which is used to replace $\sigma_t$ (from the predefined noise schedule) with a corrected $\hat{\sigma}_t$ in the denoising process. The NLC network utilizes the encoder of a pretrained denoiser network $\epsilon$.

They get -NLC versions of the DDIM, DDPM and EDM sampling algorithms for unconditional image generation, for which they conduct experiments on CIFAR10.

They also incorporate their NLC approach in algorithms for constrained sampling, specifically applied to image restoration. For this, they do experiments on super resolution, deblurring, colorization, compressed sensing and inpainting, on both ImageNet and CelebA-HQ.

Their -NLC versions of the sampling algorithms consistently improve the generated sample quality at least to some extent across all experiments.

**Audience:**

Yes

**Broader Impact Concerns:**

No concerns.

**Claims And Evidence:**

No

**Requested Changes:**

The studied problem is relevant and quite interesting. This could definitely be relevant for the TMLR audience.

However, I think that some clarifications are required in the current version of the paper, see "Weaknesses" above.

**Strengths And Weaknesses:**

Strengths:
- The studied problem is relevant and quite interesting.
- The paper is quite well written overall, most things are explained well.
- The proposed approach makes sense overall, it is conceptually quite simple.
- The proposed approach achieves at least small performance gains across all experiments.




Weaknesses:
- It is not entirely clear to me exactly how the noise level correction network is trained, I think this should be clarified a bit. In eq. (14), what is the expectation over? It is over $x_0, t, \epsilon$ as in eq. (4)? I also think it could be clarified a bit further how the objective in eq. (14) is derived.
- In particular, it is not clear to me why you need to introduce the scaling factor $\lambda$ in eq. (14)? There are no ablation results showing how this affects the performance either? Is this $\lambda$ crucial for the observed performance gains of the proposed approach?
- The additional computational cost at training and test-time of the proposed approach is only briefly mentioned at the very end of Section 4, I think this should be discussed in some more detail. In Appendix D.2, test-time cost for DDIM-NLC vs DDIM is shown, but there is no comparison of the image restoration methods in Table 3/4. How does IterProj-NLC compare with DDNM-NLC and DDNM?






Minor things:
- Section 1, "robotic path-planningand control", typo.
- Section 1, "Previous studies Rick Chang et al. (2017); Permenter & Yuan (2024) have", incorrect citation formatting.
- Section 1: "This perspective views the sampling process an optimization problem" --> "This perspective views the sampling process as an optimization problem"?
- Figure 1, "Compressive Sensing" --> "Compressed Sensing"?
- Section 1, "such as DDNM (Denoising Diffusion Nullspace Model) in Wang et al. (2023)" --> "such as DDNM (Denoising Diffusion Nullspace Model) by Wang et al. (2023)"?
- Section 2.1, "can be found in Song et al. (2021b); Karras et al. (2022)", citation formatting.
- Section 2.1, "the randomized DDPM Ho et al. (2020) follows the update rule", citation formatting.
- Section 3.1, "Previous studies Rick Chang et al. (2017); Permenter & Yuan (2024) have established this connection", citation formatting.
- Section 3.1, "Permenter & Yuan (2024) introduces" --> "Permenter & Yuan (2024) introduce"?
- First line after eq. (13), the expression of $\hat{x}_{0|t}$ should match eq. (5)?
- Section 3.1, "the imperfections of the denoiser Ning et al. (2023)", citation formatting.
- Section 3.2, "As illustrated in Permenter & Yuan (2024), this occurs": I would do either "As illustrated by Permenter & Yuan (2024), this occurs" or "As illustrated in (Permenter & Yuan, 2024), this occurs".
- Section 4.1, "Theorom 1", typo. "as shown in ??", missing reference.
- Figure 4 caption, "(b) Distance Estimation Error" --> "(b) Distance Estimation Bias"?
- 4.2, "following standard practice in image generation tasks Heusel et al. (2017)", citation formatting.
- 4.3: "RePaint ?", missing reference.
- Table 7, "Heper" --> "Hyper".

---

> ### Author Response · Authors · 2025-04-06
> **Response to Reviewer A2wi**
>
> We sincerely thank the reviewer A2wi for their detailed and constructive feedback. Below, we address each of your points with clarifications and improvements incorporated into the revised manuscript.
>
> ---
>
> ### **Q1. Clarification on the training procedure and objective in Eq. (14)**
>
> **A1.** Thank you for raising this point. We have revised Equation (14) in the main text to clearly define the expectation and variables involved.
> The training objective is to align the corrected noise level $\hat{\sigma}_t$ with the true distance from the noisy sample $x_t$ to the data manifold $\mathcal{K}$. The expectation is taken over the joint distribution of the variables:
> $x_0$ sampled from the training dataset,
>  $t$ uniformly drawn from the diffusion time steps (e.g., $t \in [1, 1000]$),
>  $\epsilon \sim \mathcal{N}(0, I)$,
>  $\lambda \sim \mathcal{U}(1 - \delta, 1 + \delta)$.
>
> We have provided a detailed explanation and derivation of the training objective in **Appendix A.2** of the revised manuscript.
> The core insight behind the proposed objective is to align the estimated noise level $ \hat{\sigma}_t \$ with the true distance from the noisy sample $ x_t$ to the data manifold, normalized by $ \sqrt{n} $. Specifically, we aim to minimize the following loss:
> $$
> \mathbb{E}[  ( \sqrt{n}  \hat{\sigma}_t - \text{dist}_K(x_t) )^2]
> $$
>
> Rather than directly predicting the distance, we adopt a residual learning strategy. We train a network to estimate the correction to the predefined noise level, defining the corrected noise level as:
> $ \hat{\sigma}_t = \sigma_t [1 + r(x_t, \sigma_t)] , $
> where $r(x_t, \sigma_t)  $ is the residual output by the noise level correction network.
>
> Since the exact distance $ \text{dist}_K(x_t) $ is generally intractable, we approximate it using the known clean counterpart $ x_0 $ from the forward diffusion process:
> $  \text{dist}_K(x_t)  \approx \| x_t - x_0 \|. $
>
> This formulation provides a practical and effective way to supervise the noise level correction network, and additional derivation details can be found in **Appendix A.2**.
>
> ---
>
> ### **Q2. Motivation and effect of the scaling factor $\lambda$**
>
> **A2.** The introduction of the scaling factor $\lambda$ in our training setup is not only for empirical convenience, but serves a key purpose: to improve the generalization of the noise level correction network $ r_\theta(x_t, \sigma_t) $.
> In diffusion sampling, cumulative denoising errors often lead to discrepancies between the actual noise level in a sample $ x_t $ and the scheduled noise level $\sigma_t $. To make the correction network robust to such mismatches, we intentionally perturb the forward diffusion process during training using a scaling factor:
> $$x_t = x_0 + \sigma_t \lambda \epsilon, \lambda \sim  \mathcal{U}(1-\delta,1+\delta)$$
>
> This introduces $\lambda$ variation in the true noise level, which becomes $ \lambda \sigma_t | \epsilon| $, while the network still conditions on the original $ \sigma_t $. This setup forces $ r_\theta $ to learn to correct residual biases under realistic mismatched conditions, making it more robust at inference time.
>
> We have clarified this motivation in the revised manuscript and added an ablation study in **Appendix D.3** to show the effect of different values of  $ \delta $ on performance. The results below demonstrate that a moderate scaling range (e.g., $ \delta = 0.5 $) improves generalization and leads to better sample quality compared to the unperturbed case $ \delta = 0 $.
>
> #### **FID Scores for Different Scaling Ranges**
>
> | $\delta$ | CIFAR-10 (DDIM-NLC) | ImageNet Inpainting (DDNM-NLC) |
> |----------|---------------------|--------------------------------|
> | 0        | 3.52                | 7.87                           |
> | 0.2      | 3.23                | 7.59                           |
> | **0.5**  | **3.12**            | **7.20**                       |
> | 1.0      | 3.14                | 7.24                           |
>
> As shown, setting $ \delta = 0.5 $ achieves the best results. We adopt this setting throughout all our experiments.

---

> > ### Author Response · Authors · 2025-04-06
> > **Cont. Response to Reviewer A2wi**
> >
> > ---
> >
> > ### **Q3. Additional computational cost**
> >
> > **A3.** The additional computational overhead introduced by the proposed NLC method is minimal. During inference, NLC adds only one extra forward pass through a lightweight residual network $r_\theta $ per denoising step. This network is significantly smaller than the main denoiser  $ \epsilon_\theta  $, resulting in a modest increase in computation.
> > Importantly, this additional step is agnostic to the sampling algorithm and can be applied to both unconstrained and constrained generation settings. We provide a detailed analysis in **Appendix D.2**, which shows that the inference overhead introduced by NLC is consistently under **6%**.
> >
> > To further validate this, we report the inference times for a real 10-step sample generation task using a batch size of 1. The experiments were conducted on two representative settings:  CIFAR-10 (unconstrained generation with DDPM), and  ImageNet (inpainting using DDNM).
> >
> > As shown below, the inference time increase from baseline to NLC-enhanced versions is approximately 6% in both cases.
> >
> > #### Table: Inference Time Comparison (in seconds)
> >
> > | Model Variant          | CIFAR-10 (DDPM) | ImageNet (DDNM-Inpainting) |
> > |------------------------|------------------|-----------------------------|
> > | Baseline               | 0.32             | 0.93                        |
> > | Baseline + NLC         | 0.34             | 0.95                        |
> >
> > These results confirm that NLC achieves improved sample quality at a minimal computational cost, making it practical for both research and deployment scenarios.
> >
> > ---
> >
> > ### **Q4. Minor issues**
> >
> > **A4.** Thank you for pointing out the typos and formatting issues. We have corrected all the minor comments in the revised manuscript.

---

> > > ### Comment · Reviewer_A2wi · 2025-04-08
> > >
> > > Thank you for the reply.
> > >
> > > I have read the other reviews and all rebuttals.
> > >
> > > All reviews are positive overall, and the authors have provided a solid rebuttal.
> > >
> > > I will recommend accept.

---

> ### Author Response · Authors · 2025-04-21
> **Response by Authors**
>
> We’re glad that our revisions have addressed your concerns, and we appreciate the time you’ve taken to help improve our work.

---

### Review · Reviewer_FJeZ · 2025-02-25

**Summary Of Contributions:**

The paper presents a simple yet effective method to enhance the denoising process by introducing a compact noise level correction (NLC) network. Despite a modest increase in training and inference costs, the proposed approach significantly improves sample generation quality.

**Audience:**

Yes

**Broader Impact Concerns:**

No concerns on the ethical implications.

**Claims And Evidence:**

Yes

**Requested Changes:**

Since this work is a modification to the existing diffusion schedulers, it is good to show the adaptability of NLC to various schedulers and models.

**Strengths And Weaknesses:**

Pros:
- The NLC network is designed as a modular, plug-and-play component, which, while incurring additional computational costs, integrates seamlessly into existing diffusion frameworks.

- The use of a lookup table approximation offers an efficiency-oriented alternative. Although this method shows a slight performance trade-off compared to the full NLC network, it remains a practical option in scenarios where computational resources are limited.

- Figure 2 clearly illustrates the advantages of the proposed noise level correction strategy. Figure 5 provides compelling evidence of the improvements during the denoising process. Experiments on several schedulers have been done to show the improvements from the NLC network.

- The method not only enhances quantitative performance in unconstrained image generation but also benefits various image restoration tasks, including super-resolution, deblurring, and colorization.

- The paper is well-organized and clearly written.

Cons:

- A question regarding the adaptability of the NLC: it remains unclear whether the noise level correction approach can be extended to consistency models or integrated with other popular schedulers such as DPM. Addressing this point could provide further insights and broaden the applicability of the method.

Overall, I think it is a solid paper to benefit the diffusion-based research works.

---

> ### Author Response · Authors · 2025-04-06
> **Response to Reviewer FjeZ**
>
> We sincerely thank Reviewer FjeZ for their thoughtful review and valuable suggestions. Below, we address your question regarding the adaptability of our proposed Noise Level Correction (NLC) method to other sampling algorithms.
>
> ---
>
> #### **Q1: Can NLC be extended to other sampling methods like Consistency Models or DPM-Solver?**
>
> **A1:** Yes, the proposed Noise Level Correction (NLC) method is designed to be broadly applicable and can be integrated into a wide range of diffusion sampling algorithms.
>
> In general, most diffusion samplers follow a two-step structure in each denoising iteration:
> 1. Estimate the clean sample $\hat{x}_{0|t}$ from the noisy observation $x_t$.
> 2. Propagate to the next time step by adding noise based on the estimated clean sample.
>
> Our NLC method fits naturally into this structure by refining the noise level used in step 1, leading to more accurate estimation of the clean sample. This integration is formalized in Equations (16) and (17) of the main paper.
> In the original submission, we demonstrated the integration of NLC into:
> - DDIM/DDPM (deterministic and stochastic samplers),
> - EDM (a second-order ODE solver), and
> - DDNM (a constrained sampler for image restoration).
>
> To further extend the generality of our method, we now include results integrating NLC with DPM-Solver, a state-of-the-art ODE-based sampler for diffusion models. This integration is described in **Algorithm 5** in **Appendix B** of the revised manuscript.
>
> We also conducted experiments on CIFAR-10 to evaluate this extension. The results in terms of FID score, reported in **Appendix D.5**, show that DPM-NLC consistently outperforms the original DPM baseline across various function evaluation (NFE) settings:
>
> | **Method**  | **NFE=12** | **NFE=18** | **NFE=24** | **NFE=30** | **NFE=36** |
> |-------------|------------|------------|------------|------------|------------|
> | DPM         | 5.28       | 3.43       | 3.02       | 2.85       | 2.78       |
> | DPM-NLC     | **4.83**   | **3.22**   | **2.97**   | **2.81**   | **2.77**   |
>
> These results confirm that NLC remains effective when combined with high-performance solvers like DPM, even in low-NFE regimes.

---

> > ### Comment · Reviewer_FJeZ · 2025-04-19
> > **Thanks for the rebuttal**
> >
> > I have read the authors' rebuttal and the other reviews. The additional experiment with DPM demonstrates the broader applicability of the method. While the improvements are somewhat limited to high-order samplers in high NFE settings, the results still offer valuable insights into the diffusion process and show promise, particularly in low NFE scenarios. The added ablation study on the parameter λ also clarifies the training procedure. My concerns have been adequately addressed.
> >
> > One additional point: I’m curious whether the authors plan to open-source their code for reproducibility. There may be some implementation-specific questions—for instance, it is not entirely clear how the noise level correction is applied to the original noise, especially given the potential mismatch in feature dimensions.

---

> > > ### Author Response · Authors · 2025-04-21
> > > **Response by Authors**
> > >
> > > We’re glad our revisions address your concerns. We will open‑source our implementation after acceptance.
> > >
> > > For further clarification, the Noise Level Correction (NLC) network takes the middle‑layer features extracted from a pretrained denoiser as input and outputs a scalar noise‑level estimate for each sample. This scalar reflects the sample’s distance from the learned image manifold and is inherently agnostic to the shape or resolution of both the noise vector and the original image. As a result, it can be directly applied in the denoising process without considering the shape of the oise vector. Further technical details are provided in Section 3.2 of the paper.

---

> > > > ### Comment · Reviewer_FJeZ · 2025-04-22
> > > > **Thanks for the answer**
> > > >
> > > > Thanks for the detailed clarification. I found Figure 3 a bit misleading — it might be helpful to present it more clearly.

---

### Review · Reviewer_iHw6 · 2025-03-24

**Summary Of Contributions:**

The paper leverages theoretical results on viewing diffusion model sampling as an optimization problem where the distance of a noisy sample from the data manifold can be approximated using the magnitude of the noise at any time t. The authors propose to correct this noise using a residual correction scheme that utilizes training an extra neural network (in addition to a pretrained denoiser) for modeling these residuals. Empirical results are demonstrated in the context of unconditional and conditional generation.

**Audience:**

Yes

**Claims And Evidence:**

Yes

**Requested Changes:**

See the Weaknesses Section above

**Strengths And Weaknesses:**

Strengths:

1. The presented technique is intuitive and can be readily applied to existing samplers in diffusion models with minimal implementation overhead.

Weaknesses / Requested Changes/Questions:

1. The authors start with Assumption 3.1 but claim this assumption might not hold in diffusion models due to error propagation during the denoising process. However, except for the visually intuitive illustration in Fig.2, no empirical evidence is provided to support this claim. I think demonstrating the fallacy of this assumption in the context of a toy example could be very useful (let's say comparing ground truth distance dist(x_t) with its approximation \sqrt{n}\sigma_t). I see the experiment in Section 4.1, but to my understanding, the presentation in Section 4.1 relatively compares DDIM with DDIM + NLC.
2. The motivation behind including the lambda parameter is unclear from the explanation in the main text. Is it primarily due to empirical convenience?
3. Re. the results on unconditional generation in Table 2, the difference between the Heuns solver in EDM and the Heun’s + NLC solver is marginal at best. Does this imply that using higher-order solvers like DPM-Solver and its variants would not benefit much from this technique?
4. In my opinion, the discussion of related work in Section 2.2 needs significant revision. The authors overlook a lot of work on improving noise schedules in diffusion models. Here are some references I found from a quick search [1,2,3], and the authors should do their own due diligence when updating this section, which would improve the paper quite a bit.
[1] Diffusion Models With Learned Adaptive Noise, Sahoo et al.
[2] Improved Noise Schedule for Diffusion Training, Hang et al.
[3] On the Importance of Noise Scheduling for Diffusion Models, Ting Chen

Minor Comments:

1. The original image row is missing from Fig. 1. Including this would make this figure better.

2. There seems to be a typo below the statement of Theorem 3.1. Notably the Tweedie’s estimate should be equal to x_0t = x_t - \sigma_t e_\theta(x_t,  t)

---

> ### Author Response · Authors · 2025-04-06
> **Response to Reviewer iHw6**
>
> We sincerely thank the reviewer for their detailed and constructive feedback. We have carefully revised the manuscript based on your suggestions.
>
> ---
>
> **Q1. Clarification on Assumption 3.1 and lack of empirical support**
>
> **A1.** In the revised manuscript, we have updated Figure 4(a) to explicitly compare the true distance to the manifold, $\text{dist}_{\mathcal{K}}(x_t)$, with the predefined noise level estimate $\sqrt{n}\sigma_t$. This comparison clearly shows that in the DDIM baseline, the estimated noise level diverges from the true distance, especially in later denoising steps, empirically demonstrating that Assumption 3.1 does not hold in practice. This supports the intuitive illustration provided in Figure 2.
>
> In contrast, the proposed DDIM-NLC method aligns the estimated noise level more closely with the true distance, resulting in samples that remain consistently closer to the manifold throughout the denoising process. Additional supporting results and discussion are provided in **Appendix C.2**.
>
> ---
>
> **Q2. Motivation behind the introduction of the $\lambda$ parameter**
>
> **A2.** The introduction of the scaling factor $\lambda$ in our training setup is not only for empirical convenience, but serves a key purpose: to improve the generalization of the noise level correction network $ r_\theta(x_t, \sigma_t) $.
> In diffusion sampling, cumulative denoising errors often lead to discrepancies between the actual noise level in a sample $ x_t $ and the scheduled noise level $\sigma_t $. To make the correction network robust to such mismatches, we intentionally perturb the forward diffusion process during training using a scaling factor:
> $$x_t = x_0 + \sigma_t \lambda \epsilon, \lambda \sim  \mathcal{U}(1-\delta,1+\delta)$$
>
> This introduces $\lambda$ variation in the true noise level, which becomes $ \lambda \sigma_t | \epsilon| $, while the network still conditions on the original $ \sigma_t $. This setup forces $ r_\theta $ to learn to correct residual biases under realistic mismatched conditions, making it more robust at inference time.
>
> We have clarified this motivation in the revised manuscript and added an ablation study in **Appendix D.3** to show the effect of different values of  $ \delta $ on performance. The results below demonstrate that a moderate scaling range (e.g., $ \delta = 0.5 $) improves generalization and leads to better sample quality compared to the unperturbed case $ \delta = 0 $.
>
> #### **FID Scores for Different Scaling Ranges**
>
> | $\delta$ | CIFAR-10 (DDIM-NLC) | ImageNet Inpainting (DDNM-NLC) |
> |----------|---------------------|--------------------------------|
> | 0        | 3.52                | 7.87                           |
> | 0.2      | 3.23                | 7.59                           |
> | **0.5**  | **3.12**            | **7.20**                       |
> | 1.0      | 3.14                | 7.24                           |
>
> As shown, setting $ \delta = 0.5 $ achieves the best results. We adopt this setting throughout all our experiments.
>
>
> ---
>
> **Q3. Marginal improvement when using high-order samplers like Heun/DPM**
>
> **A3.** The limited improvements in some higher-order sampling scenarios can be attributed to their already high performance, which leaves less room for further enhancement.
> For example, on CIFAR-10 with 35-step Heun, the FID improves from 1.98 to 1.95, a modest gain due to the strong baseline. However, at lower steps (e.g., 13-step Heun), our method improves FID from 2.71 to 2.56, showing a more substantial benefit.
>
> ---
>
> **Q4. Related work coverage is incomplete**
>
> **A4.** Thank you for the valuable references. We have substantially revised and reorganized **Section 2.2 (Related Work)** to incorporate additional categories of work, including:
> - Adaptive noise scheduling (e.g., Sahoo et al., Hang et al., Chen et al.)
> - Few-step sampling methods
> - Diffusion models for inverse problems
>
> ---
>
> **Q5. Minor comments**
>
> **A5.** Thank you. We have:
> - Corrected the typo.
> - Added the missing original image row to Figure 1, improving clarity.

---

### Decision · Action_Editor_u7zQ · 2025-05-04

**Recommendation:** Accept as is

**Comment:**

This paper focuses on improving sample generation in diffusion models. Motivated by the insight that the denoising process can be viewed as projecting noise samples onto the data manifold, the authors propose a noise level correction network to enhance denoising and demonstrate its benefits across a range of downstream tasks. Reviewers agree that the work has a clear and intuitive motivation (iHw6, FJeZ), shows improved performance across a broad range of tasks (A2wi, FJeZ), and can be readily applied to existing sampling algorithms in diffusion models (iHw6). Meanwhile, reviewers also raised concerns regarding the empirical evidence (iHw6), limited analysis of adaptability (FJeZ), and insufficient details on training settings (A2wi). The authors addressed these concerns thoroughly during the rebuttal phase, and all reviewers acknowledged that their concerns were resolved. Given the overall consensus, the AE recommends acceptance of this paper.

**Audience:**

Yes.

**Claims And Evidence:**

Yes.